# Gaussian Mixture Solvers for Diffusion Models

**Hanzhong Guo**[*1,3], **Cheng Lu**[4], **Fan Bao**[4], **Tianyu Pang**[2], **Shuicheng Yan**[2],
**Chao Du**[†2], **Chongxuan Li**[†1,3]
[1]Gaoling School of Artificial Intelligence, Renmin University of China
[2]Sea AI Lab, Singapore
[3]Beijing Key Laboratory of Big Data Management and Analysis Methods, Beijing, China
[4]Tsinghua University
{allanguo, tianyupang, yansc, duchao}@sea.com;
lucheng.lc15@gmail.com; bf19@mails.tsinghua.edu.cn;
chongxuanli@ruc.edu.cn

## Abstract

Recently, diffusion models have achieved great success in generative tasks. Sampling from diffusion models is equivalent to solving the reverse diffusion stochastic differential equations (SDEs) or the corresponding probability flow ordinary differential equations (ODEs). In comparison, SDE-based solvers can generate samples of higher quality and are suited for image translation tasks like stroke-based synthesis. During inference, however, existing SDE-based solvers are severely constrained by the efficiency-effectiveness dilemma. Our investigation suggests that this is because the Gaussian assumption in the reverse transition kernel is frequently violated (even in the case of simple mixture data) given a limited number of discretization steps. To overcome this limitation, we introduce a novel class of SDE-based solvers called *Gaussian Mixture Solvers (GMS)* for diffusion models. Our solver estimates the first three-order moments and optimizes the parameters of a Gaussian mixture transition kernel using generalized methods of moments in each step during sampling. Empirically, our solver outperforms numerous SDE-based solvers in terms of sample quality in image generation and stroke-based synthesis in various diffusion models, which validates the motivation and effectiveness of GMS. Our code is available at https://github.com/Guohanzhong/GMS.

## 1 Introduction

In recent years, deep generative models and especially (score-based) diffusion models [35, 14, 37, 18] have made remarkable progress in various domains, including image generation [15, 7], audio generation [22, 32], video generation [16], 3D object generation [31], multi-modal generation [42, 5, 12, 43], and several downstream tasks such as image translation [28, 45] and image restoration [19].

Sampling from diffusion models can be interpreted as solving the reverse-time diffusion stochastic differential equations (SDEs) or their corresponding probability flow ordinary differential equations (ODEs) [37]. SDE-based and ODE-based solvers of diffusion models have very different properties and application scenarios In particular, SDE-based solvers usually perform better when given a sufficient number of discretization steps [18]. Indeed, a recent empirical study [24] suggests that SDE-based solvers can potentially generate high-fidelity samples with realistic details and intricate semantic coherence from pre-trained large-scale text-to-image diffusion models [34]. Besides, SDE-based solvers are preferable in many downstream tasks such as stroke-based synthesis [27], image translation [45], and image manipulation [20].

---

[*]Work done during an internship at Sea AI Lab. [†]Correspondence to Chao Du and Chongxuan Li.

37th Conference on Neural Information Processing Systems (NeurIPS 2023).

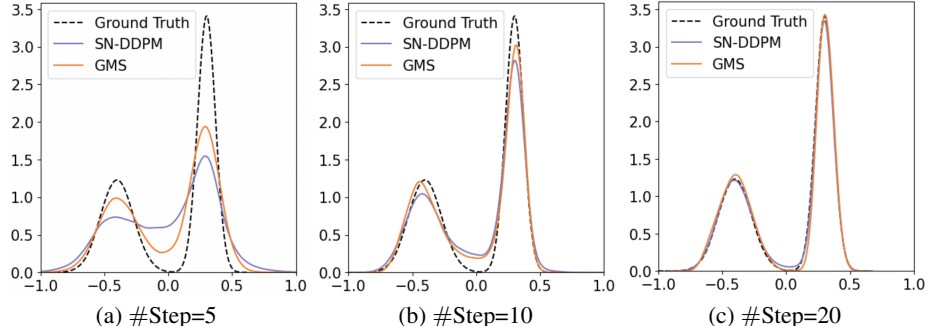

| (a) #Step=5 | (b) #Step=10 | (c) #Step=20 |

Figure 1: **Sampling on a mixture of Gaussian.** GMS (**ours**) and SN-DDPM excel in fitting the true distribution when the transition kernel is Gaussian, with a sufficient number of sampling steps (c). However, GMS outperforms SN-DDPM [2] when sampling steps are limited and the reverse transition kernel deviates from Gaussian (a-b).

Despite their wide applications, SDE-based solvers face a significant trade-off between efficiency and effectiveness during sampling, since an insufficient number of steps lead to larger discretization errors. To this end, Bao et al. [3] estimate the optimal variance of the Gaussian transition kernel in the reverse process instead of using a handcraft variance to reduce the discretization errors. Additionally, Bao et al. [2] explore the optimal diagonal variance when dealing with an imperfect noise network and demonstrate state-of-the-art (SOTA) performance with a few steps among other SDE-based solvers. Notably, these solvers all assume that the transition kernel in the reverse process is Gaussian.

In this paper, we systematically examine the assumption of the Gaussian transition kernel and reveal that it can be easily violated under a limited number of discretization steps even in the case of simple mixture data. To this end, we propose a new type of SDE-based solver called *Gaussian Mixture Solver (GMS)*, which employs a more expressive Gaussian mixture transition kernel in the reverse process for better approximation given a limited number of steps (see visualization results on toy data in Fig. 1). In particular, we first learn a noise prediction network with multiple heads that estimate the high-order moments of the true reverse transition kernel respectively. For sampling, we fit a Gaussian mixture transition kernel in each step via the *generalized methods of the moments* [11] using the predicted high-order moments of the true reverse process.

To evaluate GMS, we compare it against a variety of baselines [14, 2, 17, 36] in terms of several widely used metrics (particularly sample quality measured by FID) in the tasks of generation and stroke-based synthesis on multiple datasets. Our results show that GMS outperforms state-of-the-art SDE-based solvers [14, 2, 17] in terms of sample quality with the limited number of discretization steps (e.g., $< 100$). For instance, GMS improves the FID by 4.44 over the SOTA SDE-based solver [2] given 10 steps on CIFAR10. Furthermore, We evaluate GMS on a stroke-based synthesis task. The findings consistently reveal that GMS achieves higher levels of realism than all aforementioned SDE-based solvers as well as the widely adopted ODE-based solver DDIM [36] while maintaining comparable computation budgets and faithfulness scores (measured by $L_2$ distance). Such empirical findings validate the motivation and effectiveness of GMS.

## 2 Background

In this section, we provide a brief overview of the (score-based) diffusion models, representative SDE-based solvers for diffusion models, and applications of such solvers.

### 2.1 Diffusion models

Diffusion models gradually perturb data with a forward diffusion process and then learn to reverse such process to recover the data distribution. Formally, let $x_0 \in \mathbb{R}^n$ be a random variable with unknown data distribution $q(x_0)$. Diffusion models define the forward process $\{x_t\}_{t \in [0,1]}$ indexed by time $t$, which perturbs the data by adding Gaussian noise to $x_0$ with

$$q(x_t|x_0) = \mathcal{N}(x_t|a(t)x_0, \sigma^2(t)I). \tag{1}$$

In general, the function $a(t)$ and $\sigma(t)$ are selected so that the logarithmic signal-to-noise ratio $\log \frac{a^2(t)}{\sigma^2(t)}$ decreases monotonically with time $t$, causing the data to diffuse towards random Gaussian noise [21]. Furthermore, it has been demonstrated by Kingma et al. [21] that the following SDE shares an identical transition distribution $q_{t|0}(x_t|x_0)$ with Eq. (1):

$$\mathrm{d}x_t = f(t)x_t\mathrm{d}t + g(t)\mathrm{d}\omega, \quad x_0 \sim q(x_0), \tag{2}$$

where $\omega \in \mathbb{R}^n$ is a standard Wiener process and

$$f(t) = \frac{\mathrm{d}\log a(t)}{\mathrm{d}t}, \quad g^2(t) = \frac{\mathrm{d}\sigma^2(t)}{\mathrm{d}t} - 2\sigma^2(t)\frac{\mathrm{d}\log a(t)}{\mathrm{d}t}. \tag{3}$$

Let $q(x_t)$ be the marginal distribution of the above SDE at time $t$. Its reversal process can be described by a corresponding continuous SDE which recovers the data distribution [37]:

$$\mathrm{d}x = \left[f(t)x_t - g^2(t)\nabla_{x_t}\log q(x_t)\right]\mathrm{d}t + g(t)\mathrm{d}\bar{\omega}, \quad x_1 \sim q(x_1), \tag{4}$$

where $\bar{\omega} \in \mathbb{R}^n$ is a reverse-time standard Wiener process. The only unknown term in Eq. (4) is the score function $\nabla_{x_t}\log q(x_t)$. To estimate it, existing works [14, 37, 18] train a noise network $\epsilon_\theta(x_t, t)$ to obtain a scaled score function $\sigma(t)\nabla_{x_t}\log q(x_t)$ using denoising score matching [38]:

$$\mathcal{L}(\theta) = \int_0^1 w(t)\mathbb{E}_{q(x_0)}\mathbb{E}_{q(\epsilon)}[\|\epsilon_\theta(x_t, t) - \epsilon\|_2^2]\mathrm{d}t, \tag{5}$$

where $w(t)$ is a weighting function, $q(\epsilon)$ is standard Gaussian distribution and $x_t \sim q(x_t|x_0)$ follows Eq. (1). The optimal solution of the optimization objective Eq. (5) is $\epsilon_\theta(x_t, t) = -\sigma(t)\nabla_{x_t}\log q(x_t)$.

Hence, samples can be obtained by initiating the process with a standard Gaussian variable $x_1$, then substituting $\nabla_{x_t}\log q(x_t)$ with $-\frac{\epsilon_\theta(x_t, t)}{\sigma(t)}$ and discretizing reverse SDE Eq. (4) from $t = 1$ to $t = 0$ to generate $x_0$.

## 2.2 SDE-based solvers for diffusion models

The primary objective of SDE-based solvers lies in decreasing discretization error and therefore minimizing function evaluations required for convergence during the process of discretizing Eq. (4). Discretizing the reverse SDE in Eq. (4) is equivalent to sample from a Markov chain $p(x_{0:1}) = p(x_1)\prod_{t_{i-1}, t_i \in S_t} p(x_{t_{i-1}}|x_{t_i})$ with its trajectory $S_t = [0, t_1, t_2, ..., t_i, .., 1]$. Song et al. [37] proves that the conventional ancestral sampling technique used in the DPMs [14] that models $p(x_{t_{i-1}}|x_{t_i})$ as a Gaussian distribution, can be perceived as a first-order solver for the reverse SDE in Eq. (4). Bao et al. [3] finds that the optimal variance of $p(x_{t_{i-1}}|x_{t_i}) \sim \mathcal{N}(x_{t_{i-1}}|\mu_{t_{i-1}|t_i}, \Sigma_{t_{i-1}|t_i}(x_{t_i}))$ is

$$\Sigma_{t_{i-1}|t_i}^*(x_{t_i}) = \lambda_{t_i}^2 + \gamma_{t_i}^2 \frac{\sigma^2(t_i)}{\alpha(t_i)}\left(1 - \mathbb{E}_{q(x_{t_i})}\left[\frac{1}{d}\left\|\mathbb{E}_{q(x_0|x_{t_i})}[\epsilon_\theta(x_{t_i}, t_i)]\right\|_2^2\right]\right), \tag{6}$$

where, $\gamma_{t_i} = \sqrt{\alpha(t_{i-1})} - \sqrt{\sigma^2(t_{i-1}) - \lambda_{t_i}^2}\sqrt{\frac{\alpha(t_i)}{\sigma^2(t_i)}}$, $\lambda_{t_i}^2$ is the variance of $q(x_{t_{i-1}}|x_{t_i}, x_0)$. AnalyticDPM [3] offers a significant reduction in discretization error during sampling and achieves faster convergence with fewer steps. Moreover, SN-DDPM [2] employs a Gaussian transition kernel with an optimal diagonal covariance instead of an isotropic covariance. This approach yields improved sample quality and likelihood compared to other SDE-based solvers within a limited number of steps.

## 2.3 Applications of SDE-based solvers for stroke-based synthesis

The stroke-based image synthesis is a representative downstream task suitable for SDE-based solvers. It involves the user providing a full-resolution image $x^{(g)}$ through the manipulation of RGB pixels, referred to as the guided image. The guided image $x^{(g)}$ possibly contains three levels of guidance: a high-level guide consisting of coarse colored strokes, a mid-level guide comprising colored strokes on a real image, and a low-level guide containing image patches from a target image.

SDEdit [27] solves the task by first starting from the guided image $x^{(g)}$ and adding Gaussian noise to disturb the guided images to $x_t$ and $q(x^{(g)}(t_0)|x^{(g)}) \sim \mathcal{N}(x_{t_0}|a(t_0)x^{(g)}, \sigma^2(t)I)$ same with Eq. (1). Subsequently, it solves the corresponding reverse stochastic differential equation (SDE) up to $t = 0$ to generate the synthesized image $x(0)$ discretizing Eq. (4). Apart from the discretization steps taken by the SDE-based solver, the key hyper-parameter for SDEdit is $t_0$, the time step from which we begin the image synthesis procedure in the reverse SDE.

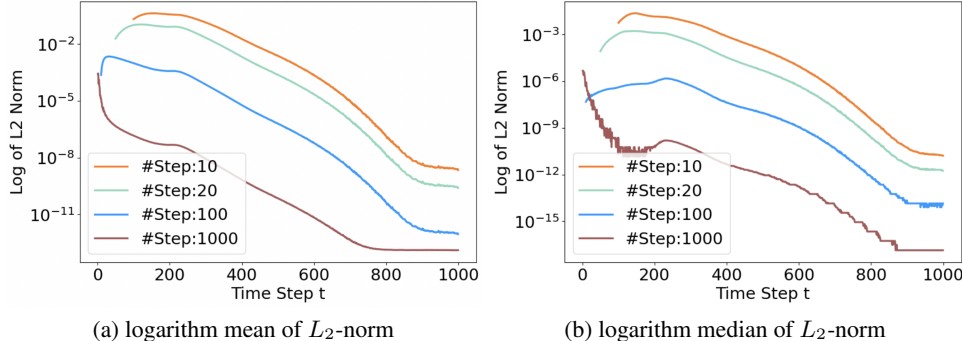

(a) logarithm mean of $L_2$-norm  (b) logarithm median of $L_2$-norm

Figure 2: **Empirical evidence of suboptimality of Gaussian kernel on CIFAR10.** (a) and (b) plot the logarithm of the image-wise mean and median of $L_2$-norm between Gaussian-assumed third-order moments and estimated third-order moments. Clearly, as the number of sampling steps decreases, the disparity between the following two third-order moments increases, denoting that the true transition kernel diverges further from the Gaussian distribution. See Appendix D.2 for more details.

## 3 Gaussian mixture solvers for diffusion models

In this section, we first show through both theoretical and empirical evidence, that the true reverse transition kernel can significantly diverge from Gaussian distributions as assumed in previous SOTA SDE-based solvers [3, 2], indicating that the reverse transition can be improved further by employing more flexible distributions (see Sec. 3.1). This motivates us to propose a novel class of SDE-based solvers, dubbed *Gaussian Mixture Solvers*, which determines a Gaussian mixture distribution via the *generalized methods of the moments* [11] using higher-order moments information for the true reverse transition (see Sec. 3.2). The higher-order moments are estimated by a noise prediction network with multiple heads on training data, as detailed in Sec. 3.3.

### 3.1 Suboptimality of Gaussian distributions for reverse transition kernels

As described in Sec. 2, existing state-of-the-art SDE-based solvers for DPMs [3, 2] approximate the reverse transition $q(x_{t_{i-1}}|x_{t_i})$ using Gaussian distributions. Such approximations work well when the number of discretization steps in these solvers is large (e.g., 1000 steps). However, for smaller discretization steps (such as when faster sampling is required), the validity of this Gaussian assumption will be largely broken. We demonstrate this theoretically and empirically below.

First, observe that $q(x_s|x_t)$[1] can be expressed as $q(x_s|x_t) = \int q(x_s|x_t, x_0)q(x_0|x_t)\mathrm{d}x_0$, which is non-Gaussian for a general data distribution $q(x_0)$. For instance, for a mixture of Gaussian or a mixture of Dirac $q(x_0)$, we can prove that the conditional distributions $q(x_s|x_t)$ in the reverse process are non-Gaussian, as characterized by Proposition 3.1, proven in Appendix A.1.

**Proposition 3.1** (Mixture Data Have non-Gaussian Reverse Kernel). *Assume $q(x_0)$ is a mixture of Dirac or a mixture of Gaussian distribution and the forward process is defined in Eq. (1). The reverse transition kernel $q(x_s|x_t), s < t$ is a Gaussian mixture instead of a Gaussian.*

Empirically, we do not know the distributions of the real data (e.g., high-dimensional images) and cannot obtain an explicit form of $q(x_s|x_t)$. However, even in such cases, it is easy to validate that $q(x_s|x_t)$ are non-Gaussian. In particular, note that the third-order moment of one Gaussian distribution ($M_3^{(G)}$) can be represented by its first-order moment ($M_1$) and its second-order moment ($M_2$) [2], which motivates us to check whether the first three order moments of $q(x_s|x_t)$ satisfy the relationship induced by the Gaussian assumption. We perform an experiment on CIFAR-10 and estimate the first three orders of moments $\hat{M}_1, \hat{M}_2, \hat{M}_3$ for $q(x_s|x_t)$ from data by high-order noise networks (see details in Sec. D.2). As shown in Fig. 2, we plot the mean and median of $l_2$-norm

---

[1]To enhance the clarity of exposition, we introduce the notation $s \doteq t_{i-1}$ and $t \doteq t_i$ to denote two adjacent time steps in the trajectory. Consequently, we refer to $q(x_s|x_t)$ as the reverse transition kernel in this context.

[2]The scalar case is $M_3^{(G)} = M_1^3 + 3M_1(M_2 - M_1^2)$. See Appendix D.2 for the vector case.

between the estimated third-order moment $\hat{M}_3$ and third-order moment $M_3^{(G)}$ calculated under the Gaussian assumption given the different number of steps at different time steps $t$. In particular, when the number of steps is large (i.e., #Step=1000), the difference between the two calculation methods is small. As the number of steps decreases, the $l_2$-norm increases, indicating that the true reverse distribution $q(x_s|x_t)$ is non-Gaussian. With the time step closer to $t = 0$, the $l_2$-norm increases too.

Both the theoretical and empirical results motivate us to weaken the Gaussian assumption in SDE-based solvers for better performance, especially when the step size is large.

## 3.2  Sampling with Gaussian mixture transition kernel

There are extensive choices of $p(x_s|x_t)$ such that it is more powerful and potentially fits $q(x_s|x_t)$ better than a Gaussian. In this paper, we choose a simple mixture of the Gaussian model as follows:

$$p(x_s|x_t) = \sum_{i=1}^{M} w_i \mathcal{N}(x_s|\mu_i(x_t), \Sigma_i(x_t)), \quad \sum_{i=1}^{M} w_i = 1, \tag{7}$$

where $w_i$ is a scalar and $\mu_i(x_t)$ and $\Sigma_i(x_t)$ are vectors. The reasons for choosing a Gaussian mixture model are three-fold. First, a Gaussian mixture model is multi-modal (e.g., see Proposition 3.1), potentially leading to a better performance than Gaussian with few steps. Second, when the number of steps is large and the Gaussian is nearly optimal [35, 3], a designed Gaussian mixture model such as our proposed kernel in Eq. (9) can degenerate to a Gaussian when the mean of two components are same, making the performance unchanged. Third, a Gaussian mixture model is relatively easy to sample. For the sake of completeness, we also discuss other distribution families in Appendix C.

Traditionally, the EM algorithm is employed for estimating the Gaussian mixture [26]. However, it is nontrivial to apply EM here because we need to learn the reverse transition kernel in Eq. (7) for all time step pairs $(s, t)$ by individual EM processes where we need to sample multiple $x_s$[3] given a $x_t$ to estimate the parameters in the Gaussian mixture indexed by $(s, t)$. This is time-consuming, especially in a high-dimensional space (e.g., natural images) and we present an efficient approach.

For improved flexibility in sampling and training, diffusion models introduce the parameterization of the noise network $\epsilon(x_t, t)$ [14] or the data prediction network $x_0(x_t, t)$ [18]. With such a network, the moments under the $q(x_s|x_t)$ measure can be decomposed into moments under the $q(x_0|x_t)$ measure so that sampling any $x_t$ to $x_s$ requires only a network whose inputs are $x_t$ and $t$, such as the decomposition shown in Eq. (10). In previous studies, Gaussian transition kernel was utilized, allowing for the distribution to be directly determined after obtaining the estimated first-order moment and handcrafted second-order moment [14] or estimated first-order and second-order moment [3]. In contrast, such a feature is not available for the Gaussian mixtures transition kernel in our paper.

Here we present the method to determine the Gaussian mixture given a set of moments and we will discuss how to obtain the moments by the parameterization of the noise network in Sec. 3.3. Assume the length of the estimated moments set is $N$ and the number of parameters in the Gaussian mixture is $d$. We adopt a popular and theoretically sound method called the generalized method of moments (GMM) [11] to learn the parameters by:

$$\min_{\theta} Q(\theta, M_1, ..., M_N) = \min_{\theta} [\frac{1}{N_c} \sum_{i=1}^{N_c} g(x_i, \theta)]^T W [\frac{1}{N_c} \sum_{i=1}^{N_c} g(x_i, \theta)], \tag{8}$$

where $\theta \in \mathbb{R}^{d \times 1}$ includes all parameters (e.g., the mean of each component) in the Gaussian mixture defined in Eq. (7). For instance, $d = 2M * D_{\text{data}} + M - 1$ in Eq. (7), where $D_{\text{data}}$ represents the number of dimensions of the data and $\theta$ contains $M$ mean vectors with $D_{\text{data}}$ dimensions, $M$ variance vectors of with $D_{\text{data}}$ dimensions (considering only the diagonal elements), and $M - 1$ weight coefficients which are scalar. The component of $g(x_i, \theta) \in \mathbb{R}^{N \times 1}$ is defined by $g_n(x_i, \theta) = M_n(x_i) - M_n^{(GM)}(\theta)$, where $M_n(x_i)$ is the $n$-th order empirical moments stated in Sec. 3.3 and $M_n^{(GM)}(\theta)$ is the $n$-th order moments of Gaussian mixture under $\theta$, and $W$ is a weighted matrix, $N_c$ is the number of samples.

Theoretically, the parameter $\hat{\theta}_{\text{GMM}}$ obtained by GMM in Eq. (8) consistently converges to the potential optimal parameters $\theta^*$ for Gaussian mixture models given the moments' condition because $\sqrt{N_c}(\hat{\theta}_{\text{GMM}} - \theta^*) \xrightarrow{d} \mathcal{N}(0, \mathbb{V}(\theta_{GMM}))$, as stated in Theorem 3.1 in Hansen [11].

---

[3]Note that $x_s$ serves as the "training sample" in the corresponding EM process.

Hence, we can employ GMM to determine a Gaussian mixture transition kernel after estimating the moments of the transition kernel. To strike a balance between computational tractability and expressive power, we specifically focus on the first three-order moments in this work and define a Gaussian mixture transition kernel with two components shown in Eq. (9) whose vectors $\mu_t^{(1)}$, $\mu_t^{(2)}$, and $\sigma_t^2$ are parameters to be optimized. This degree of simplification is acceptable in terms of its impact. Intuitively, such a selection has the potential to encompass exponential modes throughout the entire trajectory. Empirically, we consistently observe that utilizing a bimodal Gaussian mixture yields favorable outcomes across all experimental configurations.

$$p(x_s|x_t) = \frac{1}{3}\mathcal{N}(\mu_t^{(1)}, \sigma_t^2) + \frac{2}{3}\mathcal{N}(\mu_t^{(2)}, \sigma_t^2), \tag{9}$$

meanwhile, under the simplification in our paper, the number of parameters $d = 3 * D_{\text{data}}$ in our Gaussian transition kernel is equal to the number of moments condition $N = 3 * D_{\text{data}}$. According to proposition 3.2, under the selection of arbitrary weighted weights, the asymptotic mean (asymptotically consistent) and asymptotic variance of the estimator remain consistent, proof in Appendix A.2. Hence, any choice of weighted weights is optimal. To further streamline optimization, we set $W = I$.

**Proposition 3.2** (Any weighted matrix is optimal for $d = N$). *Assume the number of parameters $d$ equals the number of moments condition $N$, and $\epsilon_\theta^n(x_t, t) \xrightarrow{p} \mathbb{E}_{q(x_0|x_t)}[diag(\epsilon \otimes^{n-1} \epsilon)]$ (where $\otimes^n$ denotes n-fold outers product) which denotes n-th order noise network converging in probability. The asymptotic variance $\mathbb{V}(\theta_{GMM})$ and the convergence speed of GMM remain the same no matter which weighted matrix is adopted in Eq. (8). Namely, any weighted matrix is optimal.*

What's more, we provide a detailed discussion on the selection of parameters such as the choice of different parameters to optimize and the different choices of weight $w_i$ in the Gaussian mixture in Appendix E.1. Combining with Sec. 4.1, our empirical findings illustrate the efficacy of the GMS across various benchmarks via using the Gaussian transition kernel in Eq. (9) fitted by the objective function shown in Eq. (31) in Appendix B.1 via the ADAN [41] as optimization method. Details regarding the parameters for this optimizer are provided in Appendix E.5.

### 3.3 Estimating high-order moments for non-Gaussian reverse process

In Sec. 3.2, we have explored the methodology for determining a Gaussian mixture transition kernel given a set of moments $M_1, ..., M_n$. In the subsequent section, we will present an approach for estimating these moments utilizing noise networks and elucidate the process of network learning.

Given the forward process described by Eq. (1), it can be inferred that both $q(x_t|x_s)$ and $q(x_s|x_t, x_0)$ follow Gaussian distributions. Specifically, $q(x_t|x_s) \sim \mathcal{N}(x_t|a_{t|s}x_s, \sigma_{t|s}^2 I)$ [21], where $a_{t|s} = \frac{\alpha(t)}{\alpha(s)}$ and $\sigma_{t|s}^2 = \sigma^2(t) - a_{t|s}^2\sigma^2(s)$. Consequently, we can deduce that $\mathbb{E}_{q(x_s|x_t)}[x_s \otimes^n x_s] = \mathbb{E}_{q(x_0|x_t)q(x_s|x_t,x_0)}[x_s \otimes^n x_s]$, where $\otimes^n$ denotes $n$-fold outer product. Thus, we can first utilize the Gaussian property of $q(x_s|x_t, x_0)$ and employ an explicit formula to calculate the moments under the measure of $q(x_s|x_t, x_0)$. What's more, we only consider the diagonal elements of higher-order moments for computational efficiency, similar to Bao et al. [2] since estimating full higher-order moments results in escalated output dimensions (e.g., quadratic growth for covariance and cubic for the third-order moments) and thus requires substantial computational demands. The expression of diagonal elements of the third-order moment of the reverse transition kernel can be derived as:

$$\hat{M}_3 = \mathbb{E}_{q(x_s|x_t)}[\text{diag}(x_s \otimes x_s \otimes x_s)] = \mathbb{E}_{q(x_0|x_t)}\mathbb{E}_{q(x_s|x_t,x_0)}[\text{diag}(x_s \otimes x_s \otimes x_s)] = \tag{10}$$

$$\underbrace{[(\frac{a_{t|s}\sigma_s^2}{\sigma_t^2})^3\text{diag}(x_t \otimes x_t \otimes x_t) + 3\lambda_t^2\frac{a_{t|s}\sigma_s^2}{\sigma_t^2}x_t]}_{\text{Constant term}}$$

$$+ \underbrace{[\frac{3a_{t|s}^2\sigma_s^4 a_{s|0}^2\beta_{t|s}^2}{\sigma_t^8}(\text{diag}(x_t \otimes x_t)) + \frac{a_{s|0}\beta_{t|s}}{\sigma_t^2}I] \odot \mathbb{E}_{q(x_0|x_t)}[x_0]}_{\text{Linear term in } x_0}$$

$$+ \underbrace{3\frac{a_{t|s}\sigma_s^2}{\sigma_t^2}(\frac{a_{s|0}\beta_{t|s}}{\sigma_t^2})^2 x_t \odot \mathbb{E}_{q(x_0|x_t)}[\text{diag}(x_0 \otimes x_0)]}_{\text{Quadratic term in } x_0} + \underbrace{(\frac{a_{s|0}\beta_{t|s}}{\sigma_t^2})^3\mathbb{E}_{q(x_0|x_t)}[\text{diag}(x_0 \otimes x_0 \otimes x_0)]}_{\text{Cubic term in } x_0},$$

---

**Algorithm 1** Learning of the high order noise network

---
1: **Input:** The $n$-th order noise network $\epsilon_{\theta,\phi_n}^n(x_t, t)$ and its tunable parameter $\phi_n$
2: **repeat**
3:     $x_0 \sim q(x_0)$
4:     $t \sim \text{Uniform}([1, ..., T])$
5:     $\epsilon_t \sim \mathcal{N}(0, I)$
6:     $x_t = \alpha(t)x_0 + \sigma(t)\epsilon_t$
7:     Take a gradient step on $\phi_n^{(t)} \leftarrow \nabla_{\phi_n} \left\| \epsilon_t^n - \epsilon_{\theta,\phi_n}^n(x_t, t) \right\|_2^2$
8: **until** converged

---

**Algorithm 2** Sampling via GMS with the first three order moments

---
1: **Input:** The assembled noise network $f^3(x_t, t)$ in Eq. (13)
2: $x_T \sim \mathcal{N}(0, I)$
3: **for** $t = T$ **to** $1$ **do**
4:     $z_t \sim \mathcal{N}(0, I)$
5:     $\hat{M}_1, \hat{M}_2, \hat{M}_3 = h(f_{[1]}^3(x_t, t), f_{[2]}^3(x_t, t), f_{[3]}^3(x_t, t))$ in Appendix B
6:     $\pi_k^*, \mu_k^*, \sigma_k^* = \min_{\pi_k, \mu_k, \sigma_k} Q(\pi_k, \mu_k, \sigma_k, \hat{M}_1, \hat{M}_2, \hat{M}_3)$ in Sec. 3.2
7:     $i \sim \pi_k^*, x_{t-1} = \mu_i^* + \sigma_i^* z_t$
8: **end for**
9: **return** $x_0$

---

where $\otimes$ is the outer product $\odot$ is the Hadamard product. Additionally, $\lambda_t^2$ corresponds to the variance of $q(x_s|x_t, x_0)$, and further elaboration on this matter can be found in Appendix B.

In order to compute the $n$-th order moment $\hat{M}_n$, as exemplified by the third-order moment in Eq. (10), it is necessary to evaluate the expectations $\mathbb{E}_{q(x_0|x_t)}[x_0], \ldots, \mathbb{E}_{q(x_0|x_t)}[\text{diag}(x_0 \otimes^{n-1} x_0)]$. Furthermore, by expressing $x_0$ as $x_0 = \frac{1}{\sqrt{\alpha(t)}}(x_t - \sigma(t)\epsilon)$, we can decompose $\hat{M}_n$ into a combination of terms $\mathbb{E}_{q(x_0|x_t)}[\epsilon], \ldots, \mathbb{E}_{q(x_0|x_t)}[(\epsilon \odot^{n-1} \epsilon)]$, which are learned by neural networks. The decomposition of third-order moments $\hat{M}_3$, as outlined in Eq. (29), is provided to illustrate this concept. Therefore in training, we learn several neural networks $\{\epsilon_\theta^n\}_{n=1}^N$ by training on the following objective functions:

$$\min_{\{\epsilon_\theta^n\}_{n=1}^N} \mathbb{E}_t \mathbb{E}_{q(x_0)q(\epsilon)} \|\epsilon^n - \epsilon_\theta^n(x_t, t)\|_2^2, \tag{11}$$

where $\epsilon \sim \mathcal{N}(0, I)$, $x_t = \alpha(t)x_0 + \sigma(t)\epsilon$, and $\epsilon^n$ denotes $\epsilon \odot^{n-1} \epsilon$. After training, we can use $\{\epsilon_\theta^n\}_{n=1}^N$ to replace $\{\mathbb{E}_{q(x_0|x_t)}[\epsilon \odot^{n-1} \epsilon]\}_{n=1}^N$ and estimate the moments $\{\hat{M}_n\}_{n=1}^N$ of reverse transition kernel.

However, in the present scenario, it is necessary to infer the network a minimum of $n$ times in order to make a single step of GMS. To mitigate the high cost of the aforementioned overhead in sampling, we adopt the two-stage learning approach proposed by Bao et al. [2]. Specifically, in the first stage, we optimize the noise network $\epsilon_\theta(x_t, t)$ by minimizing the expression $\min \mathbb{E}_t \mathbb{E}_{q(x_0)q(\epsilon)} \|\epsilon - \epsilon_\theta(x_t, t)\|_2^2$ or by utilizing a pre-trained noise network as proposed by [14, 37]. In the second stage, we utilize the optimized network as the backbone and keep its parameters $\theta$ fixed, while adding additional heads to generate the $n$-th order noise network $\epsilon_{\theta,\phi_n}^n(x_t, t)$.

$$\epsilon_{\theta,\phi_n}^n(x_t, t) = \text{NN}(\epsilon_\theta(x_t, t), \phi_n), \tag{12}$$

where NN is the extra head, which is a small network such as convolution layers or small attention block, parameterized by $\phi_n$, details in Appendix E.3. We present the second stage learning procedure in Algorithm 1. Upon training all the heads, we can readily concatenate the outputs of different heads, as the backbone of the higher-order noise network is shared. By doing so, we obtain the assembled noise network $f^N(x_t, t)$. When estimating the $k$-th order moments, it suffices to extract only the first $k$ components of the assembled noise network.

$$f^N(x_t, t) = \text{concat}([\underbrace{\epsilon_\theta(x_t, t), \epsilon_{\theta,\phi_2}^2(x_t, t), .., \epsilon_{\theta,\phi_k}^k(x_t, t)}_{\text{Required for estimating } \hat{M}_k}, ..., \epsilon_{\theta,\phi_N}^N(x_t, t)]), \tag{13}$$

Table 1: **Comparison with competitive SDE-based solvers w.r.t. FID score ↓ on CIFAR10 and ImageNet 64×64.** our GMS outperforms existing SDE-based solvers in most cases. SN-DDPM denotes Extended AnalyticDPM from Bao et al. [2].

| | CIFAR10 (LS) | | | | | | | |
|---|---|---|---|---|---|---|---|---|
| # TIMESTEPS $K$ | 10 | 20 | 25 | 40 | 50 | 100 | 200 | 1000 |
| DDPM, $\tilde{\beta}_t$ | 43.14 | 25.28 | 21.63 | 15.24 | 15.21 | 10.94 | 8.23 | 5.11 |
| DDPM, $\beta_t$ | 233.41 | 168.22 | 125.05 | 82.31 | 66.28 | 31.36 | 12.96 | 3.04 |
| SN-DDPM | 21.87 | 8.32 | 6.91 | 4.99 | 4.58 | 3.74 | 3.34 | 3.71 |
| GMS (OURS) | **17.43** | **7.18** | **5.96** | **4.52** | **4.16** | **3.26** | **3.01** | **2.76** |

| | CIFAR10 (CS) | | | | | | IMAGENET $64 \times 64$ | | | | | |
|---|---|---|---|---|---|---|---|---|---|---|---|---|
| # TIMESTEPS $K$ | 10 | 25 | 50 | 100 | 200 | 1000 | 25 | 50 | 100 | 200 | 400 | 4000 |
| DDPM, $\tilde{\beta}_t$ | 34.76 | 16.18 | 11.11 | 8.38 | 6.66 | 4.92 | 29.21 | 21.71 | 19.12 | 17.81 | 17.48 | 16.55 |
| DDPM, $\beta_t$ | 205.31 | 84.71 | 37.35 | 14.81 | 5.74 | **3.34** | 170.28 | 83.86 | 45.04 | 28.39 | 21.38 | 16.38 |
| SN-DDPM | 16.33 | 6.05 | 4.19 | 3.83 | 3.72 | 4.08 | 27.58 | 20.74 | 18.04 | 16.72 | 16.37 | 16.22 |
| GMS (OURS) | **13.80** | **5.48** | **4.00** | **3.46** | **3.34** | 4.23 | **26.50** | **20.13** | **17.29** | **16.60** | **15.98** | **15.79** |

To this end, we outline the GMS sampling process in Algorithm 2, where $f_{[2]}^3(x_t, t)$ represents concat($[\epsilon_\theta(x_t, t), \epsilon_{\theta, \phi_2}^2(x_t, t)]$). In comparison to existing methods with the same network structure, we report the additional memory cost of the assembled noise network in Appendix E.6.

# 4 Experiment

In this section, we first illustrate that GMS exhibits superior sample quality compared to existing SDE-based solvers when using both linear and cosine noise schedules [14, 30]. Additionally, we evaluate various solvers in stroke-based image synthesis (i.e., SDEdit) and demonstrate that GMS surpasses other SDE-based solvers, as well as the widely used ODE-based solver DDIM [36].

## 4.1 Sample quality on image data

In this section, we conduct a quantitative comparison of sample quality using the widely adopted FID score [13]. Specifically, we evaluate multiple SDE-based solvers, including a comparison with DDPM [14] and Extended AnalyticDPM [2] (referred to as SN-DDPM) using the even trajectory.

As shown in Tab. 1, GMS demonstrates superior performance compared to DDPM and SN-DDPM under the same number of steps in CIFAR10 and ImageNet $64 \times 64$. Specifically, GMS achieves a remarkable 4.44 improvement in FID given 10 steps on CIFAR10. Appendix E.7 illustrates in more detail the improvement of GMS when the number of sampling steps is limited. Meanwhile, we conduct GMS in ImageNet $256 \times 256$ via adopting the U-ViT-Huge [4] backbone as the noise network in Appendix E.8. Furthermore, taking into account the additional time required by GMS, our method still exhibits improved performance, as detailed in Appendix E.9. For integrity, we provide a comparison with other SDE-based solvers based on continuous time diffusion such as Gotta Go Fast [17], EDM [18] and SEED [10] in Appendix E.10 and shows that GMS largely outperforms other SDE-based solvers when the number of steps is less than 100. In Appendix G, we provide generated samples from GMS.

## 4.2 Stroke-based image synthesis based on SDEdit [27]

**Evaluation metrics.** We evaluate the editing results based on realism and faithfulness similar with Meng et al. [27]. To quantify the realism of sample images, we use FID between the generated images and the target realistic image dataset. To quantify faithfulness, we report the $L_2$ distance summed over all pixels between the stroke images and the edited output images.

SDEdit [27] applies noise to the stroke image $x^g$ at time step $t_0$ using $\mathcal{N}(x_{t_0}^g | \alpha(t) x^g, \sigma^2(t) I)$ and discretize the reverse SDE in Eq. (4) for sampling. Fig. 9 demonstrates the significant impact of $t_0$ on the realism of sampled images. As $t_0$ increases, the similarity to real images decreases. we choose

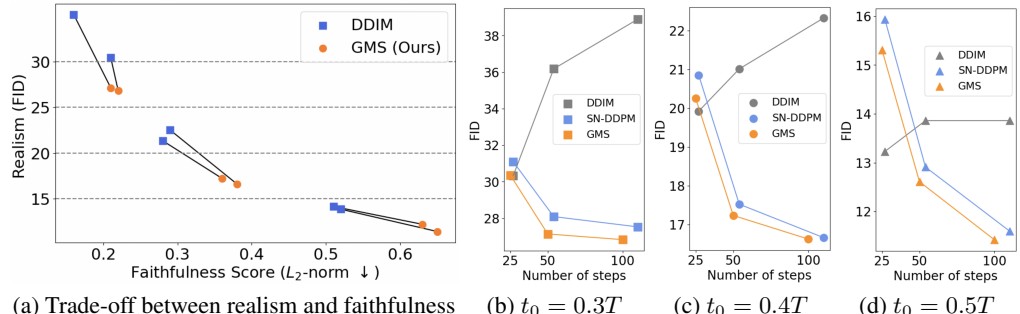

Figure 3: **Result among different solvers in SDEdit.** $t_0$ denotes the time step of the start of reverse. **(a)**: The points on each line represent the same $t_0$ and the number of sampling steps. We select $t_0 = [300, 400, 500]$, number of steps = $[50, 100]$. **(b)**: When $t_0 = 300$, the effect of DDIM diminishes more prominently with the increase in the number of steps.

the range from $t_0 = 0.3T$ to $t_0 = 0.5T$ ($T = 1000$ in our experiments) for our further experiments since sampled images closely resemble the real images in this range.

Fig. 3(a) illustrates that when using the same $t_0$ and the same number of steps, edited output images from GMS have lower faithfulness but with higher realism. This phenomenon is likely attributed to the Gaussian noise introduced by the SDE-based solver during each sampling step. This noise causes the sampling to deviate from the original image (resulting in low faithfulness) but enables the solver to transition from the stroke domain to the real image domain. Fig. 3(b) to Fig. 3(d) further demonstrates this phenomenon to a certain extent. The realism of the sampled images generated by the SDE-based solver escalates with an increase in the number of sampling steps. Conversely, the realism of the sampled images produced by the ODE-based solver diminishes due to the absence of noise, which prevents the ODE-based solver from transitioning from the stroke domain to the real image domain. Additionally, in the SDEdit task, GMS exhibits superior performance compared to SN-DDPM [2] in terms of sample computation cost. Fig. 4 shows the samples using DDIM and GMS when $t_0 = 400$ and the number of steps is 40.

## 5 Related work

**Faster solvers.** In addition to SDE-based solvers, there are works dedicated to improving the efficiency of ODE-based solvers [25, 23, 8]. Some approaches use explicit reverse transition kernels, such as those based on generative adversarial networks proposed by Xiao et al. [40] and Wang et al. [39]. Gao et al. [9] employ an energy function to model the reverse transition kernel. Zhang and Chen [44] use a flow model for the transition kernel.

**Non-Gaussian diffusion.** Apart from diffusion, some literature suggests using non-Gaussian forward processes, which consequently involve non-Gaussian reverse processes. Bansal et al. [1] introduce a generalized noise operator that incorporates noise. Nachmani et al. [29] incorporate Gaussian mixture or Gamma noise into the forward process. While these works replace both the forward and reverse processes with non-Gaussian distributions, our approach aims to identify a suitable combination of non-Gaussian distributions to model the reverse process.

## 6 Conclusion

This paper presents a novel Gaussian mixture solver (GMS) for diffusion models. GMS relaxes the Gaussian reverse kernel assumption to reduce discretization errors and improves the sample quality under the same sampling steps. Experimental results show that GMS outperforms existing SDE-based solvers, achieving a remarkable 4.44 improvement in FID compared to the state-of-the-art SDE-based solver proposed by Bao et al. [2] given 10 steps. Furthermore, due to the presence of noise, SDE-based solvers prove more suitable for stroke-based synthesis tasks and GMS still outperforms state-of-the-art SDE-based solvers.

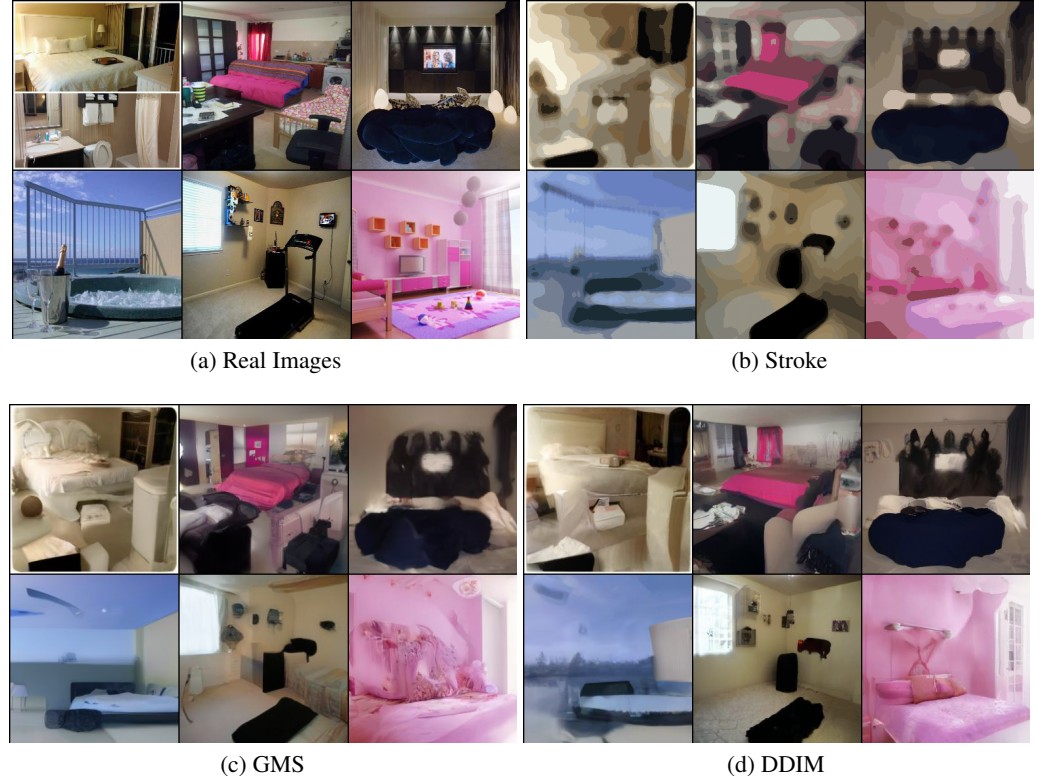

(a) Real Images          (b) Stroke

(c) GMS          (d) DDIM

Figure 4: **SDEdit samples of GMS and DDIM.** Due to the presence of noise, SDE-based solvers, such as GMS **OURS**, generate images with more details.

**Limitations and broader impacts.** While GMS enhances sample quality and potentially accelerates inference speed compared to existing SDE-based solvers, employing GMS still fall short of real-time applicability. Like other generative models, diffusion models can generate problematic fake content, and the use of GMS may amplify these undesirable effects.

## Acknowledgement

This work was supported by NSF of China (Nos. 62076145); Beijing Outstanding Young Scientist Program (No. BJJWZYJH012019100020098); Major Innovation & Planning Interdisciplinary Platform for the "Double-First Class" Initiative, Renmin University of China; the Fundamental Research Funds for the Central Universities, and the Research Funds of Renmin University of China (No. 22XNKJ13). C. Li was also sponsored by Beijing Nova Program (No. 20220484044).

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

# A Proof

## A.1 Proof of Proposition 3.1

When $q(x_0)$ is a mixture of Dirac distribution which means that $q(x_0) = \sum_{i=1}^{M} w_i \delta(x - x_i), \sum_{i=1}^{M} w_i = 1$, which has total $M$ components, and when the forward process $q(x_t|x_0)$ is a Gaussian distribution as Eq. (1), the reverse process $q(x_s|x_t)$ would be:

$$q(x_s|x_t) = \int q(x_s|x_t, x_0) q(x_0|x_t) dx_0 = \int q(x_s|x_t, x_0) q(x_0) q(x_t|x_0)/q(x_t) dx_0$$

$$= 1/q(x_t) \int q(x_s|x_t, x_0) q(x_0) q(x_t|x_0) dx_0 = 1/q(x_t) \sum_{i=1}^{M} w_i q(x_s|x_t, x_0^i) q(x_t|x_0^i)$$

According to the definition of forward process, the distribution $q(x_t|x_0^i) = \mathcal{N}(x_t|\sqrt{\bar{a}_t}x_0^i, (1 - \bar{a}_t)I)$. Due to the Markov property of forward process, when $t > s$, we have $q(x_s, x_t|x_0) q(x_s|x_0) q(x_t|x_s)$. The term $q(x_s|x_t, x_0)$ would be viewed as a Bayesian posterior resulting from a prior $q(x_s|x_0)$, updated with a likelihood term $q(x_t|x_s)$. And therefore

$$q(x_s|x_t, x_0^i) = N(x_s|\mu_q(x_t, x_0), \Sigma_q(x_t, x_0)I) = \mathcal{N}(x_s|\frac{a_{t|s}\sigma_s^2}{\sigma_t^2}x_t + \frac{a_s\sigma_{t|s}^2}{\sigma_t^2}x_0, \frac{\sigma_s^2\sigma_{t|s}^2}{\sigma_t^2}I)$$

It is easy to prove that, the distribution $q(x_s|x_t)$ is a mixture of Gaussian distribution:

$$q(x_s|x_t) \propto \sum_{i=1}^{M} w_i q(x_s|x_t, x_0^i) q(x_t|x_0^i)$$

$$= \sum_{i=1}^{M} w_i \mathcal{N}(x_t|\sqrt{\bar{a}_t}x_0^i, (1 - \bar{a}_t)I) * \mathcal{N}(x_s|\mu_q(x_t, x_0), \Sigma_q(x_t, x_0))$$

When $t$ is large, $s$ is small, $\sigma_{t|s}^2$ would be large, meaning that the influence of $x_0^i$ would be large.

Secondly, when $q(x_0)$ is a mixture of Gaussian distribution which means that $q(x_0) = \sum_{i=1}^{M} w_i \mathcal{N}(x_0^i|\mu_i, \Sigma_i), \sum_{i=1}^{M} w_i = 1$. For simplicity of analysis, we assume that this distribution is a one-dimensional distribution or that the covariance matrix is a high-dimensional Gaussian distribution with a diagonal matrix $\Sigma_i = \text{diag}_i(\sigma^2)$. Similar to the situation above, for each dimension in the reverse process:

$$q(x_s|x_t) = 1/q(x_t) \int q(x_s|x_t, x_0) q(x_0) q(x_t|x_0) dx_0$$

$$= \sum_{i=1}^{M} w_i/q(x_t) \int q(x_s|x_t, x_0) \mathcal{N}(x_0|\mu_i, \Sigma_i) q(x_t|x_0) dx_0$$

$$= \sum_{i=1}^{M} w_i/q(x_t) \int \frac{1}{\sqrt{2\pi}\sigma_q} e^{-\frac{(x_0^i - \mu_q)^2}{\sigma_q^2}} \frac{1}{\sqrt{2\pi}\sigma_i} e^{-\frac{(x_0^i - \mu_i)^2}{\sigma_i^2}} \frac{1}{\sqrt{2\pi}\sqrt{1 - \bar{a}_t}} e^{-\frac{(x_t - \sqrt{\bar{a}_t}x_0)^2}{1 - \bar{a}_t}} dx_0$$

$$= \sum_{i=1}^{M} w_i/q(x_t) \int Z_i \frac{1}{\sqrt{2\pi}\sigma_x} e^{-\frac{(x_0^i - \mu_x)^2}{\sigma_x^2}} e^{-\frac{(x_s - \mu(x_t))^2}{\sigma(x_t)}} dx_0$$

And $q(x_s|x_t)$ could be a Gaussian mixture which has $M$ component.

## A.2 Proof of Proposition 3.2

Recall that our objective function is to find optimal parameters:

$$\hat{\theta} = \underset{\theta}{\text{argmin}}[\underbrace{Q_{N_c}(\theta)}_{1 \times 1}] = \underset{\theta}{\text{argmin}}[\underbrace{g_{N_c}(\theta)^T}_{1 \times N} \underbrace{W_{N_c}}_{N \times N} \underbrace{g_{N_c}(\theta)}_{1 \times N}] \tag{14}$$

where $g_M(\theta)$ is the moment conditions talked about in Sec. 3.2, $W$ is the weighted Matrix, $N_c$ is the sample size. When solving such problems using optimizers, which are equivalent to the one we used when selecting $\theta_{GMM}$ such that $\frac{\partial Q_T(\theta_{GMM})}{\partial\theta} = 0$, and its first derivative:

$$\underbrace{\frac{\partial Q_{N_c}(\theta)}{\partial\theta}}_{d\times 1} = \begin{pmatrix} \frac{\partial Q_{N_c}(\theta)}{\partial\theta_1} \\ \frac{\partial Q_{N_c}(\theta)}{\partial\theta_2} \\ \frac{\partial Q_{N_c}(\theta)}{\partial\theta_3} \end{pmatrix}, \quad \frac{\partial Q_{N_c}(\theta)}{\partial\theta_m} = 2\underbrace{[\frac{1}{N_c}\sum_{i=1}^{N_c}\frac{\partial g(x_i,\theta)}{\partial\theta_m}]^T}_{1\times N}\underbrace{W_{N_c}}_{N\times N}\underbrace{[\frac{1}{N_c}\sum_{i=1}^{M}g(x_i,\theta)]}_{N\times 1} \quad (15)$$

its second derivative (the Hessian matrix):

$$\underbrace{\frac{\partial^2 Q_{N_c}(\theta)}{\partial\theta^2}}_{d\times d} = \begin{pmatrix} \frac{\partial^2 Q_{N_c}(\theta)}{\partial\theta_1\theta_1} & \frac{\partial^2 Q_{N_c}(\theta)}{\partial\theta_1\theta_2} & \cdots & \frac{\partial^2 Q_{N_c}(\theta)}{\partial\theta_1\theta_d} \\ \frac{\partial^2 Q_{N_c}(\theta)}{\partial\theta_2\theta_1} & \cdots & \cdots & \cdots \\ \cdots & \cdots & \cdots & \cdots \\ \cdots & \cdots & \cdots & \frac{\partial^2 Q_{N_c}(\theta)}{\partial\theta_d\theta_d} \end{pmatrix} \quad (16)$$

$$, \frac{\partial^2 Q_{N_c}(\theta)}{\partial\theta_i\theta_j} = 2[\frac{1}{M}\sum_{i=1}^{N_c}\frac{\partial g(x_i,\theta)}{\partial\theta_i}]^T W_{N_c}[\frac{1}{N_c}\sum_{i=1}^{N_c}\frac{\partial g(x_i,\theta)}{\partial\theta_j}]$$
$$+ 2[\frac{1}{N_c}\sum_{i=1}^{N_c}\frac{\partial^2 g(x_i,\theta)}{\partial\theta_i\theta_j}]W_{N_c}[\frac{1}{N_c}\sum_{i=1}^{N_c}g(x_i,\theta)].$$

By Taylor's expansion of the gradient around optimal parameters $\theta_0$, we have:

$$\frac{\partial Q_{N_c}(\theta_{GMM})}{\partial\theta} - \frac{\partial Q_{N_c}(\theta_0)}{\partial\theta} \approx \frac{\partial^2 Q_{N_c}(\theta_0)}{\partial\theta\partial\theta^T}(\theta_{GMM} - \theta) \quad (17)$$

$$\longmapsto (\theta_{GMM} - \theta) \approx -(\frac{\partial^2 Q_{N_c}(\theta_0)}{\partial\theta\partial\theta^T})^{-1}\frac{\partial Q_{N_c}(\theta_0)}{\partial\theta}.$$

Consider one element of the gradient vector $\frac{\partial Q_{N_c}(\theta_0)}{\partial\theta_m}$

$$\frac{\partial Q_{N_c}(\theta_0)}{\partial\theta_m} = 2\underbrace{[\frac{1}{M}\sum_{i=1}^{N_c}\frac{\partial g(x_i,\theta_0)}{\partial\theta_m}]^T}_{\xrightarrow{P}\mathbb{E}(\frac{\partial g(x_i,\theta_0)}{\partial\theta_m})=\Gamma_{0,m}}\underbrace{W_{N_c}}_{\xrightarrow{P}W}\underbrace{[\frac{1}{N_c}\sum_{i=1}^{N_c}g(x_i,\theta_0)]}_{\xrightarrow{P}\mathbb{E}[g(x_i,\theta_0)]=0}. \quad (18)$$

Consider one element of the Hessian matrix $\frac{\partial^2 Q_{N_c}(\theta_0)}{\partial\theta_i\partial\theta_j^T}$

$$\frac{\partial^2 Q_{N_c}(\theta_0)}{\partial\theta_i\partial\theta_j^T} = 2[\frac{1}{N_c}\sum_{i=1}^{N_c}\frac{\partial g(x_i,\theta_0)}{\partial\theta_i}]^T W_{N_c}[\frac{1}{N_c}\sum_{i=1}^{N_c}\frac{\partial g(x_i,\theta_0)}{\partial\theta_j}] \quad (19)$$
$$+ 2[\frac{1}{N_c}\sum_{i=1}^{N_c}\frac{\partial^2 g(x_i,\theta_0)}{\partial\theta_i\theta_j}]^T W_{N_c}[\frac{1}{N_c}\sum_{i=1}^{N_c}g(x_i,\theta_0)] \xrightarrow{p} 2\Gamma_{0,i}^T W\Gamma_{0,j}.$$

Therefore, it is easy to prove that $\theta_{GMM} - \theta_0 \xrightarrow{p} 0$, and uses law of large numbers we could obtain,

$$\sqrt{T}\frac{\partial Q_T(\theta_0)}{\partial\theta_m} = 2\underbrace{[\frac{1}{M}\sum_{i=1}^{M}\frac{\partial g(x_i,\theta_0)}{\partial\theta_m}]^T}_{\xrightarrow{P}\mathbb{E}(\frac{\partial g(X,\theta_0)}{\partial\theta_m})=\Gamma_{0,m}}\underbrace{W_M}_{\xrightarrow{P}W}[\underbrace{\frac{1}{\sqrt{T}}\sum_{i=1}^{M}g(x_i,\theta_0)}_{\xrightarrow{d}\mathcal{N}(0,\mathbb{E}(g(X,\theta_0)g(X,\theta_0)^T))}] \xrightarrow{d} 2\Gamma_{0,m}^T W\mathcal{N}(0,\Phi_0),$$

$$(20)$$

therefore, we have

$$\sqrt{T}(\theta_{GMM} - \theta_0) \approx -(\frac{\partial^2 Q_T(\theta_0)}{\partial\theta\partial\theta^T})^{-1}\sqrt{T}\frac{\partial Q_T(\theta_0)}{\partial\theta} \quad (21)$$
$$\xrightarrow{d} \mathcal{N}(0, (\Gamma_0^T W\Gamma_0)^{-1}\Gamma_0^T W\Phi_0 W\Gamma_0(\Gamma_0^T W\Gamma_0)^{-1}).$$

When the number of parameters $d$ equals the number of moment conditions $N$, $\Gamma_0$ becomes a (nonsingular) square matrix, and therefore,

$$
\begin{aligned}
\sqrt{T}(\theta_{GMM} - \theta_0) &\xrightarrow{d} \mathcal{N}(0, (\Gamma_0^T W \Gamma_0)^{-1} \Gamma_0^T W \Phi_0 W \Gamma_0 (\Gamma_0^T W \Gamma_0)^{-1}) \\
&= \mathcal{N}(0, \Gamma_0^{-1} W^{-1} (\Gamma_0^T)^{-1} \Gamma_0^T W \Phi_0 W \Gamma_0 \Gamma_0^{-1} W^{-1} (\Gamma_0^T)^{-1}) \\
&= \mathcal{N}(0, \Gamma_0^{-1} \Phi_0 (\Gamma_0^T)^{-1}),
\end{aligned}
\tag{22}
$$

which means that the foregoing observation suggests that the selection of $W_{N_c}$ has no bearing on the asymptotic variance of the GMM estimator. Consequently, it implies that regardless of the specific method employed to determine $W_{N_c}$, provided the moment estimates are asymptotically consistent, $W_{N_c}$ serves as the optimal weight matrix, even when dealing with small samples.

The moments utilized for fitting via the generalized method of moments in each step are computed based on the noise network's first $n$-th order. To ensure adherence to the aforementioned proposition, it is necessary to assume that the $n$-th order of the noise network converges with probability to $\mathbb{E}_{q(x_0|x_t)}[\text{diag}(\epsilon \otimes^{n-1} \epsilon)]$, details in Appendix B. Consequently, the $n$-th order moments derived from the noise networks converge with probability to the true moments. Therefore, any choice of weight matrix is optimal.

### A.3 Non-Gaussian distribution of transition kernel within large discretization steps

we can apply Bayes' rule to the posterior distribution $q(x_t|x_{t+\Delta t})$ as follows:

$$
q(x_t|x_{t+\Delta t}) = \frac{q(x_{t+\Delta t}|x_t)q(x_t)}{q(x_{t+\Delta t})} = q(x_{t+\Delta t}|x_t)\exp(\log(q(x_t)) - \log(q(x_{t+\Delta t}))) \tag{23}
$$

$$
\propto \exp\left(-\frac{\|x_{t+\Delta t} - x_t - f_t(x_t)\Delta t\|^2}{2g_t^2 \Delta t} + \log p(x_t) - \log(x_{t+\Delta t})\right),
$$

where $\Delta t$ is the step size, $q(x_t)$ is the marginal distribution of $x_t$. When $x_{t+\Delta t}$ and $x_t$ are close enough, using Taylor expansion for $\log p(x_{t+\Delta t})$, we could obtain:

$$
\log p(x_{t+\Delta t}) \approx \log p(x_t) + (x_{t+\Delta t} - x_t)\nabla_{x_t} \log p(x_t) + \Delta t \frac{\partial}{\partial t} \log p(x_t), \tag{24}
$$

$$
q(x_t|x_{t+\Delta t}) \propto \exp\left(-\frac{\left\|x_{t+\Delta t} - x_t - [f_t(x_t) - g_t^2 \nabla_{x_t} \log p(x_t)]\Delta t\right\|^2}{2g_t^2 \Delta t} + O(\Delta t)\right). \tag{25}
$$

By ignoring the higher order terms, the reverse transition kernel will be Gaussian distribution. However, as $\Delta t$ increases, the higher-order terms in the Taylor expansion cannot be disregarded, which causes the reverse transition kernel to deviate from a Gaussian distribution.

Empirically, from Fig. 2 in our paper, we observe that as $T$ decreases, the reverse transition kernel increasingly deviates from a Gaussian distribution. For more details, please refer to Appendix D.2.

## B Calculation of the first order moment and higher order moments

Suppose the forward process is a Gaussian distribution same with the Eq. (1) as $q(x_t|x_s) = N(x_t|a(t)x_{t-1}, \sigma(t)I)$.

And let $1 \geq t > s \geq 0$ always satisfy, $q(x_t|x_s) = N(x_t|a_{t|s}x_s, \beta_{t|s}I)$, where $a_{t|s} = a_t/a_s$ and $\beta_{t|s} = \sigma_t^2 - a_{t|s}^2 \sigma_s^2$, $\sigma_s = \sqrt{1 - a(s)}, \sigma_t = \sqrt{1 - a(t)}$. It's easy to prove that the distribution $q(x_t|x_0)$, $q(x_s|x_t, x_0)$ are also a Gaussian distribution [21]. Therefore, the mean of $x_s$ under the measure $q(x_s|x_t)$ would be

$$
\begin{aligned}
\mathbb{E}_{q(x_s|x_t)}[x_s] &= \mathbb{E}_{q(x_0|x_t)} \mathbb{E}_{q(x_s|x_t,x_0)}[x_s] \\
&= \mathbb{E}_{q(x_0|x_t)}\left[\frac{1}{a_{t|s}}\left(x_t - \frac{\beta_{t|s}}{\sigma_t}\epsilon_t\right)\right] \\
&= \frac{1}{a_{t|s}}\left(x_t - \frac{\beta_{t|s}}{\sigma_t}\mathbb{E}_{q(x_0|x_t)}[\epsilon_t]\right).
\end{aligned}
\tag{26}
$$

And for the second order central moment $\text{Cov}_{q(x_s|x_t)}[x_s]$, we use the total variance theorem, refer to [2], and similar with [2], we only consider the diagonal covariance.

$$\text{Cov}_{q(x_s|x_t)}[x_s] = \mathbb{E}_{q(x_0|x_t)}\text{Cov}_{q(x_s|x_t,x_0)}[x_s] + \text{Cov}_{q(x_0|x_t)}\mathbb{E}_{q(x_s|x_t,x_0)}[x_s] \tag{27}$$

$$= \lambda_t^2 I + \text{Cov}_{q(x_0|x_t)}\tilde{\mu}(x_n, \mathbb{E}_{q(x_0|x_n)}[x_0])$$

$$= \lambda_t^2 I + \frac{a_s\beta_{t|s}^2}{\sigma_t^4}\text{Cov}_{q(x_0|x_t)}[x_0]$$

$$= \lambda_t^2 I + \frac{a_{s|0}\beta_{t|s}^2}{\sigma_t^4}\frac{\sigma_t^2}{a_{t|0}}\text{Cov}_{q(x_0|x_t)}[\epsilon_t]$$

$$= \lambda_t^2 I + \frac{\beta_{t|s}^2}{\sigma_t^2 a_{t|s}}(\mathbb{E}_{q(x_0|x_t)}[\epsilon_t \odot \epsilon_t] - \mathbb{E}_{q(x_0|x_t)}[\epsilon_t] \odot \mathbb{E}_{q(x_0|x_t)}[\epsilon_t]),$$

since the higher-order moments are diagonal matrix, we use $\text{diag}(M)$ to represent the diagonal elements that have the same dimensions as the first-order moments, such as $\text{diag}(x_s \otimes x_s) = \text{Cov}_{q(x_s|x_t)}[x_s]$ and $\text{diag}(x_s \otimes x_s \otimes x_s) = \hat{M}_3$ have the same dimensions as $x_s$

Moreover, for the diagonal elements of the third-order moments, we have $\mathbb{E}_{q(x_s|x_t)}[\text{diag}(x_s \otimes x_s \otimes x_s)] = \mathbb{E}_{q(x_0|x_t)}\mathbb{E}_{q(x_s|x_t,x_0)}[x_s \odot x_s \odot x_s]$, we could use the fact that $\mathbb{E}_{q(x_s|x_t,x_0)}[(x_s - \mu(x_t,x_0)) \odot (x_s - \mu(x_t,x_0)) \odot (x_s - \mu(x_t,x_0))] = 0$, therefore,

$$\hat{M}_3 = \mathbb{E}_{q(x_s|x_t)}[\text{diag}(x_s \otimes x_s \otimes x_s)] = \mathbb{E}_{q(x_0|x_t)}\mathbb{E}_{q(x_s|x_t,x_0)}[\text{diag}(x_s \otimes x_s \otimes x_s)] \tag{28}$$

$$= \mathbb{E}_{q(x_0|x_t)}\mathbb{E}_{q(x_s|x_t,x_0)}[3\text{diag}(x_t \otimes x_t \otimes \mu(x_t,x_0))$$

$$-3\text{diag}(x_t \otimes \mu(x_t,x_0) \otimes \mu(x_t,x_0)) + \text{diag}(\mu(x_t,x_0) \otimes^2 \mu(x_t,x_0))]$$

$$= \underbrace{[(\frac{a_{t|s}\sigma_s^2}{\sigma_t^2})^3\text{diag}(x_t \otimes x_t \otimes x_t) + 3\lambda_t^2\frac{a_{t|s}\sigma_s^2}{\sigma_t^2}x_t]}_{\text{Constant term}}$$

$$+ \underbrace{[\frac{3a_{t|s}^2\sigma_s^4 a_{s|0}^2\beta_{t|s}^2}{\sigma_t^8}(\text{diag}(x_t \otimes x_t)) + \frac{a_{s|0}\beta_{t|s}}{\sigma_t^2}I] \odot \mathbb{E}_{q(x_0|x_t)}[x_0]}_{\text{Linear term in } x_0}$$

$$+ \underbrace{3\frac{a_{t|s}\sigma_s^2}{\sigma_t^2}(\frac{a_{s|0}\beta_{t|s}}{\sigma_t^2})^2 x_t \odot \mathbb{E}_{q(x_0|x_t)}[\text{diag}(x_0 \otimes x_0)]}_{\text{Quadratic term in } x_0} + \underbrace{(\frac{a_{s|0}\beta_{t|s}}{\sigma_t^2})^3\mathbb{E}_{q(x_0|x_t)}[\text{diag}(x_0 \otimes x_0 \otimes x_0)]}_{\text{Cubic term in } x_0},$$

where the third equation is derived from the decomposition of higher-order moments of Gaussian distribution, the fourth equation is obtained by splitting It is noted to highlight that in our study, we only consider the diagonal higher-order moments in our method for computational efficiency since estimating full higher-order moments results in escalated output dimensions (e.g., quadratic growth for covariance and cubic for the third-order moments) and thus requires substantial computational demand and therefore all outer products can be transformed into corresponding element multiplications and we have:

$$\mathbb{E}_{q(x_0|x_t)}[\text{diag}(x_0 \otimes^2 x_0)] = \frac{1}{\alpha^{\frac{3}{2}}(t)}\mathbb{E}_{q(x_0|x_t)}[\text{diag}((x_t - \sigma(t)\epsilon) \otimes^2 (x_t - \sigma(t)\epsilon))] \tag{29}$$

$$= \frac{1}{\alpha^{\frac{3}{2}}(t)}\mathbb{E}_{q(x_0|x_t)}[\text{diag}(x_t \otimes^2 x_t - 3\sigma(t)(x_t \otimes x_t) \otimes \epsilon$$

$$+ 3\sigma^2(t)x_t \otimes (\epsilon \otimes \epsilon) - \sigma^3(t)(\epsilon \otimes^2 \epsilon))]$$

$$= \frac{1}{\alpha^{\frac{3}{2}}(t)}[\mathbb{E}_{q(x_0|x_t)}[x_t \odot^2 x_t] - 3\sigma(t)(x_t \odot x_t) \odot \mathbb{E}_{q(x_0|x_t)}[\epsilon]+$$

$$+ 3\sigma^2(t)x_t\mathbb{E}_{q(x_0|x_t)}[\epsilon \odot \epsilon] - \sigma^3(t)[\epsilon \odot^2 \epsilon]],$$

Therefore, when we need to calculate the third-order moment, we only need to obtain $\mathbb{E}_{q(x_0|x_t)}[\epsilon_t]$, $\mathbb{E}_{q(x_0|x_t)}[\epsilon_t \odot \epsilon_t]$ and $\mathbb{E}_{q(x_0|x_t)}[\epsilon_t \odot \epsilon_t \odot \epsilon_t]$. Similarly, when we need to calculate

the $n$-order moment, we will use $\mathbb{E}_{q(x_0|x_t)}[\epsilon_t], ..., \mathbb{E}_{q(x_0|x_t)}[\epsilon_t \odot^{n-1} \epsilon_t]$. Bao et al. [2] put forward using a sharing network and using the MSE loss to estimate the network to obtain the above information about different orders of noise.

The function $h(f^3_{[1]}, f^3_{[2]}, f^3_{[3]})$ in Algo. 2 is defined as $h(f^3_{[1]}(x_t, t), f^3_{[2]}(x_t, t), f^3_{[3]}(x_t, t)) = M_1(f^3_{[1]}(x_t, t)), M_2(f^3_{[2]}(x_t, t)), M_3(f^3_{[3]}(x_t, t))$, where $M_1(f^3_{[1]}(x_t, t)) = \mathbb{E}_{q(x_s|x_t)}[x_s]$ in Eq. (26), $M_2(f^3_{[2]}(x_t, t)) = \mathrm{Cov}_{q(x_s|x_t)}[x_s]$ in Eq. (27) and $M_3(f^3_{[3]}(x_t, t)) = \hat{M}_3 = \mathbb{E}_{q(x_s|x_t)}[\mathrm{diag}(x_s \otimes x_s \otimes x_s)]$ in Eq. (28)

### B.1 Objective function for Gaussian mixture transition kernel with two components

Recall that the general objective function to fit a Gaussian mixture transition kernel in each sampling step via GMM is shown in Eq. (30).

$$\min_\theta Q(\theta, M_1, ..., M_N) = \min_\theta [\frac{1}{N_c} \sum_{i=1}^{N_c} g(x_i, \theta)]^T W [\frac{1}{N_c} \sum_{i=1}^{N_c} g(x_i, \theta)], \tag{30}$$

where $A^T$ denotes the transpose matrix of matrix A. In our paper, we propose to use the first three moments to fit a Gaussian mixture transition kernel $p(x_s|x_t) = \frac{1}{3}\mathcal{N}(\mu_t^{(1)}, \sigma_t^2) + \frac{2}{3}\mathcal{N}(\mu_t^{(2)}, \sigma_t^2)$ in each sampling step from $x_t$ to $x_s$. Therefore, the final objective function as follow:

$$\min_{\mu_t^{(1)}, \mu_t^{(2)}, \sigma_t^2} [\begin{pmatrix} M_1 - (\frac{1}{3}\mu_t^{(1)} + \frac{2}{3}\mu_t^{(2)}) \\ M_2 - (\frac{1}{3}[(\mu_t^{(1)})^2 + \sigma_t^2] + \frac{2}{3}[(\mu_t^{(2)})^2 + \sigma_t^2]) \\ M_3 - (\frac{1}{3}[(\mu_t^{(1)})^3 + 3\mu_t^{(1)}\sigma_t^2] + \frac{2}{3}[(\mu_t^{(2)})^3 + 3\mu_t^{(2)}\sigma_t^2]) \end{pmatrix}]^T I \tag{31}$$
$$[\begin{pmatrix} M_1 - (\frac{1}{3}\mu_t^{(1)} + \frac{2}{3}\mu_t^{(2)}) \\ M_2 - (\frac{1}{3}[(\mu_t^{(1)})^2 + \sigma_t^2] + \frac{2}{3}[(\mu_t^{(2)})^2 + \sigma_t^2]) \\ M_3 - (\frac{1}{3}[(\mu_t^{(1)})^3 + 3\mu_t^{(1)}\sigma_t^2] + \frac{2}{3}[(\mu_t^{(2)})^3 + 3\mu_t^{(2)}\sigma_t^2]), \end{pmatrix}]$$

where to simplify the analysis, we use the scalar form of parameters $\mu_t^{(1)}, \mu_t^{(2)}, \sigma_t^2$ as representation.

## C Modeling reverse transition kernel via exponential family

Analysis in Sec. 3.1 figures out that modeling reverse transition kernel via Gaussian distribution is no longer sufficient in fast sampling scenarios. In addition to directly proposing the use of a Gaussian Mixture for modeling, we also analyze in principle whether there are potentially more suitable distributions i.e., the feasibility of using them for modeling.

We would turn back to analyzing the original objective function of DPMs to find a suitable distribution. The forward process $q(x_t|x_s) = N(x_t|a_{t|s}x_s, \beta_{t|s}I)$, consistent with the definition in Appendix B. DPMs' goal is to optimize the modeled reverse process parameters to maximize the variational bound $L$ in Ho et al. [14]. And the ELBO in Ho et al. [14] can be re-written to the following formula:

$$L = D_{\mathrm{KL}}(q(x_T)||p(x_T)) + \mathbb{E}_q[\sum_{t \geq 1} D_{\mathrm{KL}}(q(x_s|x_t)||p(x_s|x_t))] + H(x_0), \tag{32}$$

where $q_t \doteq q(x_t)$ is the true distribution and $p_t \doteq p(x_t)$ is the modeled distribution, and the minimum problem could be transformed into a sub-problem, proved in Bao et al. [3]:

$$\min_{\{\theta\}} L \Leftrightarrow \min_{\{\theta_{s|t}\}_{t=1}^T} D_{\mathrm{KL}}(q(x_s|x_t)||p_{\theta_{s|t}}(x_s|x_t)). \tag{33}$$

We have no additional information besides when the reverse transition kernel is not Gaussian. But Lemma. C.3 proves that when the reverse transition kernel $p_{\theta_{s|t}}(x_s|x_t)$ is exponential family $p_{\theta_t}(x_s|x_t) = p(x_t, \theta_{s|t}) = h(x_t)\exp\left(\theta_{s|t}^T t(x_t) - \alpha(\theta_{s|t})\right)$, solving the sub-problem Eq. (33) equals to solve the following equations, which is to match moments between the modeled distribution and true distribution:

$$\mathbb{E}_{q(x_s|x_t)}[t(x_s)] = \mathbb{E}_{p(x_t, \theta_{s|t})}[t(x_s)]. \tag{34}$$

When $t(x) = (x, .., x^n)^T$, solving Eq.(34) equals to match the moments of true distribution and modeled distribution.

Meanwhile, Gaussian distribution belongs to the exponential family with $t(x) = (x, x^2)^T$ and $\theta_t = (\frac{\mu_t}{\sigma_t^2}, \frac{-1}{2\sigma_t^2})^T$, details in Lemma. C.2. Therefore, when modeling the reverse transition kernel as Gaussian distribution, the optimal parameters are that make its first two moments equal to the true first two moments of the real reverse transition kernel $q(x_s|x_t)$, which is consistent with the results in Bao et al. [3] and Bao et al. [2].

The aforementioned discussion serves as a motivation to acquire higher-order moments and identify a corresponding exponential family, which surpasses the Gaussian distribution in terms of complexity. However, proposition C.1 shows that finding such exponential family distribution with higher-order moments is impossible.

**Proposition C.1** (Infeasibility of exponential family with higher-order moments.). *Given the first $n$-th order moments. It's non-trivial to find an exponential family distribution for $\min D_{\mathrm{KL}}(q||p)$ when $n$ is odd. And it's hard to solve $\min D_{\mathrm{KL}}(q||p)$ when $n$ is even.*

## C.1   Proof of Proposition C.1

**Lemma C.2.** *(Gaussian Distribution belongs to Exponential Family). Gaussian distribution $p(x) = \frac{1}{\sqrt{2\pi}\sigma} \exp\left(-\frac{(x-\mu)^2}{2\sigma^2}\right)$ is exponential family with $t(x) = (x, x^2)^T$ and $\theta = (\frac{\mu}{\sigma^2}, -\frac{1}{2\sigma^2})^T$*

*Proof.* For simplicity, we only prove one-dimensional Gaussian distribution. We could obtain:

$$
\begin{aligned}
p(x) &= \frac{1}{\sqrt{2\pi}\sigma} \exp\left(-\frac{(x-\mu)^2}{2\sigma^2}\right) \\
&= \frac{1}{\sqrt{2\pi\sigma^2}} \exp\left(-\frac{1}{2\sigma^2}(x^2 - 2\mu x + \mu^2)\right) \\
&= \exp\left(\log(2\pi\sigma^2)^{-1/2}\right) \exp\left(-\frac{1}{2\sigma^2}(x^2 - 2\mu x) - \frac{\mu^2}{\sigma^2}\right) \\
&= \exp\left(\log(2\pi\sigma^2)^{-1/2}\right) \exp\left(-\frac{1}{2\sigma^2}(-2\mu \quad 1)(x \quad x^2)^T - \frac{\mu^2}{\sigma^2}\right) \\
&= \exp\left((\frac{\mu}{\sigma^2} \quad \frac{-1}{2\sigma^2})(x \quad x^2)^T - (\frac{\mu^2}{2\sigma^2} + \frac{1}{2}\log(2\pi\sigma^2))\right),
\end{aligned}
\tag{35}
$$

where $\theta = (\frac{\mu}{\sigma^2}, \frac{-1}{2\sigma^2})^T$ and $t(x) = (x, x^2)^T$ □

**Lemma C.3.** *(The Solution for Exponential Family in Minimizing the KL Divergence). Suppose that $p(x)$ belongs to exponential family $p(x, \theta) = h(x) \exp\left(\theta^T t(x) - \alpha(\theta)\right)$, and the solution for minimizing the $E_q[\log p]$ is $E_q[t(x)] = E_{p(x,\theta)}[t(x)]$.*

*Proof.* An exponential family $p(x, \eta) = h(x) \exp\left(\eta^T t(x) - \alpha(\eta)\right) \propto f(x, \eta) = h(x) \exp\left(\eta^T t(x)\right)$ with log-partition $\alpha(\eta)$. And we could obtain its first order condition on $E_q[\log p]$ as:

$$
\nabla_\eta \log f(x, \eta) = \nabla_\eta(\log h(x) + \eta^T t(x)) = t(x)
\tag{36}
$$

$$
\begin{aligned}
\nabla_\eta \alpha(\eta) &= \nabla_\eta \log\left(\int f(x, \eta)dx\right) = \frac{\int \nabla_\eta f(x, \eta)dx}{\int f(x, \eta)dx} \\
&= e^{-\alpha(\eta)} \int t(x)f(x, \eta)dx = \int t(x)p(x, \eta)dx = \mathbb{E}_{p(x,\eta)}[t(x)]
\end{aligned}
\tag{37}
$$

In order to minimize the $D_{\mathrm{KL}}(q||p) = \int q \log(q/p) = -\mathbb{E}_q[\log p]$, we have:

$$\mathbb{E}_q[\log p] = \int dq \log(h(x)) + \int dq(\eta^T t(x) - \alpha(\eta))$$

$$\implies \frac{\partial}{\partial \eta} \mathbb{E}_q[\log p] = \int dq [\frac{\partial}{\partial \eta}(\eta^T x - \alpha(\eta))] = 0$$

$$\implies \int dq(x - \mathbb{E}_{p(x,\eta)}[t(x)]) = \mathbb{E}_q[t(x)] - \mathbb{E}_{p(x,\eta)}[t(x)] = 0$$

$$\implies \mathbb{E}_q[t(x)] = \mathbb{E}_{p(x,\eta)}[t(x)]$$

For the second-order condition, we have the following:

$$\frac{\partial^2}{\partial \eta^2} \alpha(\eta) = \frac{\partial}{\partial \eta} \int dp(x,\eta) t(x) \tag{38}$$

$$= \int \frac{\partial}{\partial \eta} h(x) \exp\big(\eta^T t(x) - \alpha(\eta)\big) t(x) dx$$

$$= \int h(x) t(x) \frac{\partial}{\partial \eta} \exp\big(\eta^T t(x) - \alpha(\eta)\big) dx$$

$$= \int h(x) t(x) \exp\big(\eta^T t(x) - \alpha(\eta)\big) dx (t(x) - \mathbb{E}_p[t(x)])$$

$$= \int p(x,\eta) dx (t^2(x) - t(x) \mathbb{E}_p[t(x)])$$

$$= \mathbb{E}_{p(x,\eta)}[t^2(x)] - \mathbb{E}_{p(x,\eta)}[t(x)]^2 = Cov_{p(x,\eta)}[t(x)] \geq 0$$

Therefore, the second-order condition for the cross entropy would be:

$$\frac{\partial^2}{\partial \eta^2} \mathbb{E}_q[\log p] = \frac{\partial}{\partial \eta}(\mathbb{E}_q[t(x)] - \mathbb{E}_{p(x,\eta)}[t(x)]) \tag{39}$$

$$= -\int \frac{\partial}{\partial \eta} p(x,\eta) t(x) dx$$

$$= -\frac{\partial^2}{\partial \eta^2} \alpha(\eta) = -Cov_{p(x,\eta)}[t(x)] \leq 0$$

When we assume that the reverse process is Gaussian, the solution to Eq. (33) equals to match the moment of true distribution and modeled distribution $\mu = E_q[x]$, $\Sigma = Cov_q[x]$. $\qquad \square$

**Lemma C.4.** *(Infeasibility of the exponential family with higher-order moments). Suppose given the first $N$-th order moments $M_i, i = 1, .., N$ and modeled $p$ as an exponential family. It is nontrivial to solve the minimum problem $E_q[\log p]$ when $N$ is odd and it's difficult to solve when $N$ is even.*

*Proof.* While given the mean, covariance, and skewness of the data distribution, assume that we could find an exponential family that minimizes the KL divergence, so that the distribution would satisfy the following form:

$$L(p, \hat{\lambda}) = D_{\mathrm{KL}}(q||p) - \hat{\lambda}^T(\int pt - m) \Rightarrow \frac{\partial}{\partial p} L(p, \hat{\lambda}) = \log \frac{p(x)}{h(x)} + 1 - \hat{\lambda}^T t = 0 \tag{40}$$

$$\Rightarrow p(x) = h(x) \exp\big(\hat{\lambda}^T t - 1\big)$$

where, $t(x) = (x, x^2, x^3)$, $p = h(x) \exp\big(\lambda_0 + \lambda_1 x + \lambda_2 x^2 + \lambda_3 x^3\big)$ and $\int dp x^3 = M_3$. However, when $\lambda_3$ is not zero, $\int p = \infty$ and density can't be normalized. The situation would be the same given an odd-order moment.

Similarly, given a more fourth-order moment, we could derive that $\lambda_3 = 0$ above, and we should solve an equation $\int dp x^4 = M_4$ and $p = h(x) \exp\big(\lambda_0 + \lambda_1 x + \lambda_2 x^2 + \lambda_4 x^4\big)$. Consider such function:

$$Z(\lambda) = \int_{-\infty}^{\infty} dx \exp\big(-x^2 - \lambda x^4\big), \lambda > 0 \tag{41}$$

When $\lambda \longrightarrow 0$, we could obtain $\lim_{\lambda \to 0} Z(\lambda) = \sqrt{\pi}$ For other cases, the lambda can be expanded and then integrated term by term, which gives $Z(\lambda) \sim \sum_{n=0}^{\infty} \frac{(-\lambda)^n}{n!} \Gamma(2n + 1/2)$, but this function However, the radius of convergence of this level is 0, so when the $\lambda$ takes other values, we need to propose a reasonable expression for the expansion after the analytic extension. Therefore, for solving the equation $\int dp x^4 = M_4$, there is no analytical solution first, and the numerical solution also brings a large computational effort. $\qquad\square$

## D  More information about Fig. 1 and Fig. 2

### D.1  Experiment in Toy-data

To illustrate the effectiveness of our method, we first compare the results of different solvers on one-dimensional data.

The distribution of our toy-data is $q(x_0) = 0.4\mathcal{N}(-0.4, 0.12^2) + 0.6\mathcal{N}(0.3, 0.05^2)$ and we define our solvers in each step as $p(x_s|x_t) = \frac{1}{3}\mathcal{N}(\mu_t^{(1)}, \sigma_t^2) + \frac{2}{3}\mathcal{N}(\mu_t^{(1)}, \sigma_t^2)$ with vectors $\mu_t^{(1)}, \mu_t^{(2)}$ and $\sigma_t^2$, which can not overfit the ground truth.

We then train second and third-order noise networks on the one-dimensional Gaussian mixture whose density is multi-modal. We use a simple MLP neural network with Swish activation [33].

Moreover, we experiment with our solvers in 8-Gaussian. The result is shown in Tab. 2. GMS outperforms Extended AnalyticDPM (SN-DDPM) [2] as presented in Tab. 2, with a bandwidth of $1.05\sigma L^{-0.25}$, where $\sigma$ is the standard deviation of data and $L$ is the number of samples.

Table 2: **Comparison with SN-DDPM w.r.t. Likelihood $\mathbb{E}_q[\log p_\theta(x)] \uparrow$ on 8-Gaussian.** GMS outperforms SN-DDPM.

|  | 8-GAUSSIAN | | | |
|---|---|---|---|---|
| # $K$ | 5 | 10 | 20 | 40 |
| SN-DDPM | -0.7885 | 0.0661 | 0.0258 | 0.1083 |
| GMS | **-0.6304** | **0.0035** | **0.0624** | **0.1127** |

### D.2  Experiment in Fig. 2

In this section, we will provide a comprehensive explanation of the procedures involved in computing the discrepancy between two third-order moment calculation methods, as depicted in Fig. 2.

The essence of the calculation lies in the assumption that the reverse transition kernel follows a Gaussian distribution. By employing the following equations (considering only the diagonal elements of higher-order moments), we can compute the third-order moment using the first two-order moments:

$$\mathbb{E}_{q(x_{t_{i-1}}|x_{t_i})}[x_{t_{i-1}} \odot x_{t_{i-1}} \odot x_{t_{i-1}}]_{\text{G}} \doteq M_G = \mu \odot \mu \odot \mu + 3\mu \odot \Sigma, \tag{42}$$

where $\mu$ is the first-order moment and $\Sigma$ is the diagonal elements of second order moment, which can be calculated by the Eq. (26) and Eq. (27). Meanwhile, we can calculate the estimated third-order moment $\hat{M}_3$ by Eq. (28).

We use the pre-trained noise network from Ho et al. [14] and the second-order noise network form Bao et al. [2] and train the third-order noise network in CIFAR10 with the linear noise schedule.

Given that all higher-order moments possess the same dimension as the first-order moment $\mu$, we can directly compare the disparity between different third-order moment calculation methods using the Mean Squared Error (MSE).

Thus, to quantify the divergence between the reverse transition kernel $q(x_s|x_t)$ and the Gaussian distribution, we can utilize the following equation:

$$\text{D}_{s|t} = \log\Big(\mathbb{E}_{q(x_s|x_t)}[x_s \odot x_s \odot x_s]_{\text{G}} - \hat{M}_3\Big)^2, \tag{43}$$

where $\hat{M}_3$ is obtained via Eq. (28), and we can start at different time step $t$ and choose a corresponding $s$ to calculate the $\text{D}_{s|t}$ and draw different time step and step size $t - s$ and we can derive Fig. 2.

# E  Experimental details

## E.1  More discussion on weight of Gaussian mixture

From Proposition 3.2, we know that when the number of parameters in the Gaussian mixture equals the number of moment conditions, any choice of weight matrix is optimal. Therefore, we will discuss the choice of parameters to optimize in this section. As we have opted for a Gaussian mixture with two components $q(x_s|x_t) = \omega_1 \mathcal{N}(\mu_{s|t}^{(1)}, \Sigma_{s|t}^{(1)}) + \omega_2 \mathcal{N}(\mu_{s|t}^{(2)}, \Sigma_{s|t}^{(2)})$ as our foundational solvers, there exist five parameters (considered scalar, with the vector cases being analogous) available for optimization.

Our primary focus is on optimizing the mean and variance of the two components, as optimizing the weight term would require solving the equation multiple times. Additionally, we have a specific requirement that our Gaussian mixture can converge to a Gaussian distribution at the conclusion of optimization, particularly when the ground truth corresponds to a Gaussian distribution. In Tab. 3, we show the result of different choices of parameters in the Gaussian mixture.

Table 3: Results among different parameters in CIFAR10 (LS), the number of steps is 50. The weight of Gaussian mixture is $\omega_1 = \frac{1}{3}$ and $\omega_2 = \frac{2}{3}$

|  | $\mu_{s|t}^{(1)}, \mu_{s|t}^{(2)}, \Sigma_{s|t}$ | $\mu_{s|t}^{(1)}, \Sigma_{s|t}^{(1)}, \Sigma_{s|t}^{(2)}$ | $\mu_{s|t}, \Sigma_{s|t}^{(1)}, \Sigma_{s|t}^{(2)}$ |
|---|---|---|---|
| CIFAR10 (LS) | **4.17** | 10.12 | 4.22 |

When a parameter is not accompanied by a superscript, it implies that both components share the same value for that parameter. On the other hand, if a parameter is associated with a superscript, and only one moment contains that superscript, it signifies that the other moment directly adopts the true value for that parameter.

It is evident that the optimization of the mean value holds greater significance. Therefore, our subsequent choices for optimization are primarily based on the first set of parameters $\mu_{s|t}^{(1)}, \mu_{s|t}^{(2)}, \Sigma_{s|t}$. Another crucial parameter to consider is the selection of weights $\omega_i$. In Tab. 4, we show the result while changing the weight of the Gaussian mixture and the set of weight $\omega_1 = \frac{1}{3}, \omega_2 = \frac{1}{2}$ performs best among different weight.

Table 4: Results among different weight choices in CIFAR10 (LS), the number of steps is 50.

|  | $\omega_1 = \frac{1}{100}, \omega_2 = \frac{99}{100}$ | $\omega_1 = \frac{1}{5}, \omega_2 = \frac{4}{5}$ | $\omega_1 = \frac{1}{3}, \omega_2 = \frac{2}{3}$ | $\omega_1 = \frac{1}{2}, \omega_2 = \frac{1}{2}$ |
|---|---|---|---|---|
| CIFAR10 (LS) | 4.63 | 4.20 | **4.17** | 4.26 |

What's more, we observe that such choice of weights $(\frac{1}{3}, \frac{1}{2})$ consistently yielded superior performance among different datasets. Identifying an optimal weight value remains a promising direction for further enhancing the GMS.

## E.2  Details of pre-trained noise networks

In Tab. 5, we list details of pre-trained noise prediction networks used in our experiments.

Table 5: Details of noise prediction networks used in our experiments. LS means the linear schedule of $\sigma(t)$ [14] in the forward process of discrete time step (see Eq. (1)). CS means the cosine schedule of $\sigma(t)$ [30] in the forward process of discrete timesteps (see Eq. (1)).

|  | # TIMESTEPS $N$ | NOISE SCHEDULE | OPTIMIZER FOR GMM |
|---|---|---|---|
| CIFAR10 (LS) | 1000 | LS | ADAN |
| CIFAR10 (CS) | 1000 | CS | ADAN |
| IMAGENET 64x64 | 4000 | CS | ADAN |

### E.3 Details of the structure of the extra head

In Tab. 6, we list structure details of $NN_1$, $NN_2$ and $NN_3$ of prediction networks used in our experiments.

Table 6: $NN_1$ represents noise prediction networks and $NN_2$, $NN_3$ represent networks for estimating the second- and the third-order of noise, which used in our experiments. Conv denotes the convolution layer. Res denotes the residual block. None denotes using the original network without additional parameters

|  | $NN_1$ | $NN_2$ (NOISE) | $NN_3$ (NOISE) |
|---|---|---|---|
| CIFAR10 (LS) | NONE | CONV | RES+CONV |
| CIFAR10 (CS) | NONE | CONV | RES+CONV |
| IMAGENET 64x64 | NONE | RES+CONV | RES+CONV |

### E.4 Training Details

We use a similar training setting to the noise prediction network in [30] and [2]. On all datasets, we use the ADAN optimizer [41] with a learning rate of $10^{-4}$; we train 2M iterations in total for a higher order of noise network; we use an exponential moving average (EMA) with a rate of 0.9999. We use a batch size of 64 on ImageNet 64X64 and 128 on CIFAR10. We save a checkpoint every 50K iterations and select the models with the best FID on 50k generated samples. Training one noise network on CIFAR10 takes about 100 hours on one A100. Training on ImageNet 64x64 takes about 150 hours on one A100.

### E.5 Details of Parameters of Optimizer in Sampling

In Tab. 7, we list details of the learning rate, learning rate schedule, and warm-up steps for different experiments.

Table 7: Details of Parameters of Optimizer used in our experiments. lr Schedule means the learning rate schedule. min lr means the minimum learning rate while using the learning rate schedule, $\iota_t$ is a function with the second order growth function of sampling steps $t$.

|  | LEARNING RATE | LR SCHEDULE | MIN LR | WARM-UP STEPS |
|---|---|---|---|---|
| CIFAR10 | MAX(0.16-$\iota_t$*0.16,0.12) | COS | 0.1 | 18 |
| IMAGENET 64×64 | MAX(0.1-$\iota_t$*0.1,0.06) | COS | 0.04 | 18 |

where COS represents the cosine learning rate schedule [6]. We find that the cosine learning rate schedule works best. The cos learning rate could be formulated as follows:

$$
\alpha_{i+1} =
\begin{cases}
\frac{i}{I_w}\alpha_i & \text{if} \quad i \le I_w \\[2mm]
\max((0.5\cos\left(\frac{i-I_w}{I-I_w}\pi\right) + 1)\alpha_t, \alpha_{\min}) & \text{else}
\end{cases}
\tag{44}
$$

where, $a_t$ is the learning rate after $t$ steps, $I_w$ is the warm-up steps, $\alpha_{\min}$ is the minimum learning rate, $I$ is the total steps.

### E.6 Details of memory and time cost

In Tab. 8, we list the memory of models (with the corresponding methods) used in our experiments. The extra memory cost higher-order noise prediction network is negligible.

Table 8: Model size (MB) for different models. The model of SNDDPM denotes the model that would predict noise and the square of noise; The model of GMDDPM denotes the model that would predict noise, the square of noise, and the third power of noise.

|  | NOISE PREDICTION NETWORK (ALL BASELINES) | NOISE & SN PREDICTION NETWORKS SNDDPM | NOISE & SN PREDICTION NETWORKS (GMDDPM) |
|---|---|---|---|
| CIFAR10 (LS) | 50.11 MB | 50.11 MB | 50.52 MB (+0.8%) |
| CIFAR10 (CS) | 50.11 MB | 50.11 MB | 50.52 MB (+0.8%) |
| IMAGENET 64×64 | 115.46 | 115.87 MB | 116.28 (+0.7%) |

In Fig. 5, we present a comprehensive breakdown of the time of optimization process within GMS at each sampling step. Subfigure (a) in Fig. 5 illustrates that when the batch size is set to 100, the time of optimizing approximately 320 steps to fit a Gaussian mixture transition kernel at each step is equivalent to the time needed for one network inference, which means that the additional time of GMS for each sampling steps is about 10% when the number of optimizing steps is 30.

Meanwhile, Subfigure (a) in Fig. 5 elucidates the relation between sampling time and the number of sampling steps for both GMS and SN-DDPM. It is noteworthy that the optimization steps employed in GMS remain fixed at 25 per sampling step, consistent with our setting in experiments. Evidently, as the number of sampling steps escalates, GMS demonstrates a proportional increase in computational overhead, consistently maintaining this overhead within a 10% margin of the original computational cost.

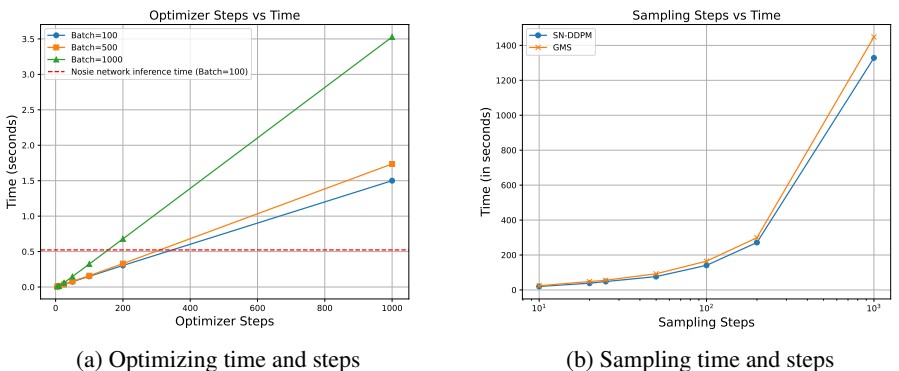

(a) Optimizing time and steps     (b) Sampling time and steps

Figure 5: **Sampling/optimizing steps and time (in seconds).** (a). The correlation between the number of optimizing steps and the corresponding optimizing time for GMS. Notably, the optimizing. (b). The correlation between the number of sampling steps and the corresponding sampling time for a single batch is observed. Notably, the sampling time demonstrates nearly linear growth with the increase in the number of sampling steps for both solvers.

Since many parts in the GMS introduce additional computational effort, Fig. 6 provides detailed information on the additional computational time of the GMS and SN-DDPM relative to the DDPM, assuming that the inference time of the noise network is unit one.

Emphasizing that the extra time is primarily for reference, most pixels can work well with a Gaussian distribution. By applying a threshold and optimizing only when the disparity between the reverse transition kernel and Gaussian exceeds it, we can conserve about 4% of computational resources without compromising result quality.

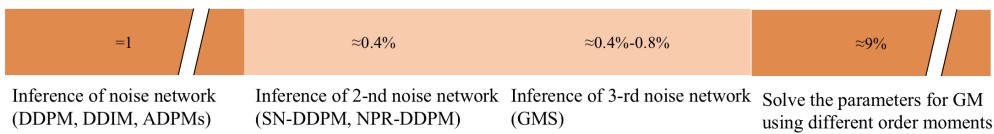

Figure 6: Time cost distribution for GMS.

### E.7 The reduction in the FID on CIFAR10 for GMS compared to SN-DDPM

To provide a more intuitive representation of the improvement of GMS compared to SN-DDPM at different sampling steps, we have constructed Fig. 7 below. As observed from Fig. 7, GMS exhibits a more pronounced improvement when the number of sampling steps is limited. However, as the number of sampling steps increases, the improvement of GMS diminishes, aligning with the hypothesis of our non-Gaussian reverse transition kernel as stated in the main text. However, we respectively clarify that due to the nonlinear nature of the FID metric, the relative/absolute FID improvements are not directly comparable across different numbers of steps

### E.8 Additional results with latent diffusion

we conduct an experiment on ImageNet $256 \times 256$ with the backbone noise network as U-ViT-Huge [4]. We train extra the second-order and the third-order noise prediction heads with two transformer blocks with the frozen backbone. The training iterations for each head is 150K and other training parameters are the same as the training of the backbone of U-ViT-Huge (details in [4]). We will evaluate the sampling efficiency of GMS under conditional sampling (classifier-free guidance scale is 0.4) and unconditional sampling in Tab. 9

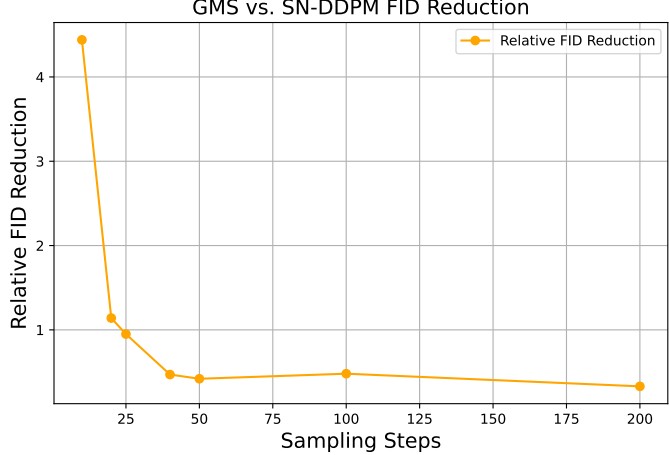

Figure 7: The reduction in the FID on CIFAR10 for GMS compared to SN-DDPM. With a restricted number of sampling steps, the improvement exhibited by GMS surpasses that achieved with an ample number of sampling steps.

Table 9: **Comparison with DDPM and SN-DDPM w.r.t. FID score ↓ on ImageNet 256 × 256 by using latent diffusion.** Uncond denotes the unconditional sampling, and Cond denotes sampling combining 90% conditional sampling with classifier-free guidance equals 0.4 and 10% unconditional sampling, consistent with Bao et al. [4].

| | COND, CFG=0.4 | | | | | | | UNCOND | | | |
|---|---|---|---|---|---|---|---|---|---|---|---|
| # TIMESTEPS $K$ | 15 | 20 | 25 | 40 | 50 | 100 | 200 | 25 | 40 | 50 | 100 |
| DDPM, $\tilde{\beta}_t$ | 6.48 | 5.30 | 4.86 | 4.42 | 4.27 | 3.93 | 3.32 | 8.62 | 6.47 | 5.97 | 5.04 |
| SN-DDPM | 4.40 | 3.36 | 3.10 | 2.99 | 2.97 | 2.93 | 2.82 | 8.19 | 5.73 | 5.32 | 4.60 |
| GMS (OURS) | **4.01** | **3.07** | **2.89** | **2.88** | **2.85** | **2.81** | **2.74** | **7.78** | **5.42** | **5.03** | **4.45** |

## E.9 Additional results with same calculation cost

Since GMS will cost more computation in the process of fitting the Gaussian mixture, we use the maximum amount of computation required (i.e., an additional 10% of computation is needed) for comparison, and for a fair comparison, we let the other solvers take 10% more sampling steps. Our GMS still outperforms existing SDE-based solvers with the same (maximum) computation cost in Tab. 10.

Table 10: **Fair comparison with competitive SDE-based solvers w.r.t. FID score ↓ on CIFAR10.** SN-DDPM denotes Extended AnalyticDPM from [2]. The number of sampling steps for the GMS is indicated within parentheses, while for other solvers, it is represented outside of parentheses.

| | CIFAR10 (LS) | | | | | | | |
|---|---|---|---|---|---|---|---|---|
| # TIMESTEPS $K$ | 11(10) | 22(20) | 28(25) | 44(40) | 55(50) | 110(100) | 220(200) | 1000(1000) |
| SN-DDPM* | 17.56 | 7.74 | 6.76 | 4.81 | 4.23 | 3.60 | 3.20 | 3.65 |
| GMS (OURS) | **17.43** | **7.18** | **5.96** | **4.52** | **4.16** | **3.26** | **3.01** | **2.76** |

In order to provide a more intuitive representation for comparing different methods under the same sampling time, we have generated Fig. 8. Subfig. (a) of Fig. 8 illustrates that at each step, there is an additional computational time cost incurred for GMS to fit the Gaussian mixture transition kernel. This computational overhead exhibits a nearly linear growth pattern with the increase in the number of sampling steps.

Subfig. (b) of Fig. 8 offers a valuable perspective on the connection between approximate sampling time and sampling quality for two GMS and SN-DDPM. It becomes apparent that GMS consistently exhibits superior performance when compared to SN-DDPM with identical computational resources. Furthermore, the data reveals that the magnitude of improvement introduced by GMS is more substantial when the number of sampling steps is limited, as opposed to scenarios with a higher number of sampling steps.

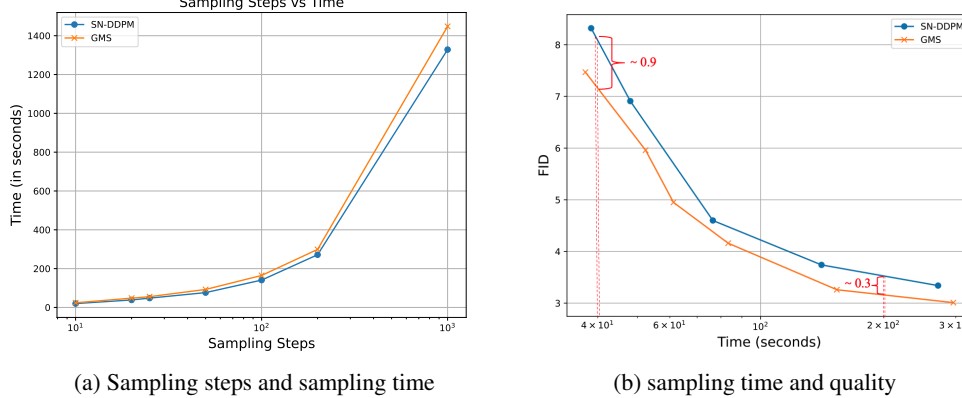

(a) Sampling steps and sampling time      (b) sampling time and quality

Figure 8: (a). The correlation between the number of sampling steps and the corresponding sampling time for a single batch is observed. Notably, the sampling time demonstrates nearly linear growth with the increase in the number of sampling steps for both solvers.(b).The relation between the sample quality (in FID) and the sampling time (in seconds) of GMS and SN-DDPM on CIFAR10.)

### E.10 Compare with continuous time SDE-based solvers

For completeness, we compare the sampling speed of GMS and non-improved reverse transition kernel in Tab. 11, and it can be seen that within 100 steps, our method outperforms other SDE-based solvers. It is worth noting that the EDM-SDE [18] is based on the $x_0$ prediction network, while SEED [10] and GMS are based on Ho et al. [14]'s pre-trained model which is a discrete-time noise network.

Table 11: **Comparison with SDE-based solvers [18, 10] w.r.t. FID score ↓ on CIFAR10.**

|  | CIFAR10 | | | | |
| --- | --- | --- | --- | --- | --- |
| # TIMESTEPS $K$ | 10 | 20 | 40 | 50 | 100 |
| SEED-2 [10] | 481.09 | 305.88 | 51.41 | 11.10 | 3.19 |
| SEED-3 [10] | 483.04 | 462.61 | 247.44 | 62.62 | 3.53 |
| EDM-SDE [18] | 35.07 | 14.04 | 9.56 | 5.12 | **2.99** |
| GMS (OURS) | **17.43** | **7.18** | **4.52** | **4.16** | 3.26 |

### E.11 Codes and License

In Tab.12, we list the code we used and the license.

Table 12: codes and license.

| URL | CITATION | LICENSE |
| --- | --- | --- |
| HTTPS://GITHUB.COM/W86763777/PYTORCH-DDPM | HO ET AL. [14] | WTFPL |
| HTTPS://GITHUB.COM/OPENAI/IMPROVED-DIFFUSION | NICHOL AND DHARIWAL [30] | MIT |

## F SDEdit

Fig. 9 illustrates one of the comprehensive procedures of SDEdit. Given a guided image, SDEdit initially introduces noise at the level of $\sigma_{t_0}$ to the original image $x_0$, where $\sigma$ denotes the noise schedules of forward process in diffusion models. Subsequently, using this noisy image $x_{t_0}$ and then discretizes it via the reverse SDE to generate the final image. Fig. 9 shows that the choice of $t_0$ will

can greatly will greatly affect the realism of sample images. With the increase of $t_0$, the similarity between sample images and the real image is decreasing. Hence, apart from conducting quantitative evaluations to assess the fidelity of the generated images, it is also crucial to undertake qualitative evaluations to examine the outcomes associated with different levels of fidelity. Taking all factors into comprehensive consideration, we have selected the range of $t_0$ from 0.3T to 0.5T in our experiments.

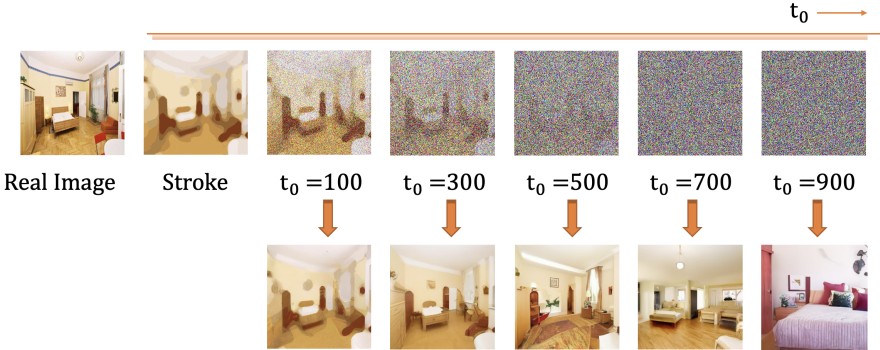

Figure 9: $t_0$ denotes the timestep to noise the stroke

Besides experiments on LSUN 256×256, we also carried out an experiment of SDEdit on ImageNet 64×64. In Tab. 13, we show the FID score for different methods in different $t_0$ and different sample steps. Our method outperforms other SDE-based solvers as well.

Table 13: **Comparison with competitive methods in SDEdit w.r.t. FID score ↓ on ImageNet 64×64.** ODE-based solver is worse than all SDE-based solvers. With nearly the same computation cost, our GMS outperforms existing methods in most cases.

| # K | IMAGENET 64X64, $t_0 = 1200$ | | | |
|---|---|---|---|---|
| | 26(28) | 51(55) | 101(111) | 201(221) |
| DDPM,$\tilde{\beta}_n$ | 21.37 | 19.15 | 18.85 | 18.15 |
| DDIM | 21.87 | 21.81 | 21.95 | 21.90 |
| SN-DDPM | 20.76 | 18.67 | 17.50 | 16.88 |
| GMS | **20.50** | **18.37** | **17.18** | **16.83** |

# G   Samples

From Fig. 10 to Fig. 12, we show generated samples of GMS under a different number of steps in CIFAR10 and ImageNet 64×64. Here we use $K$ to denote the number of steps for sampling.

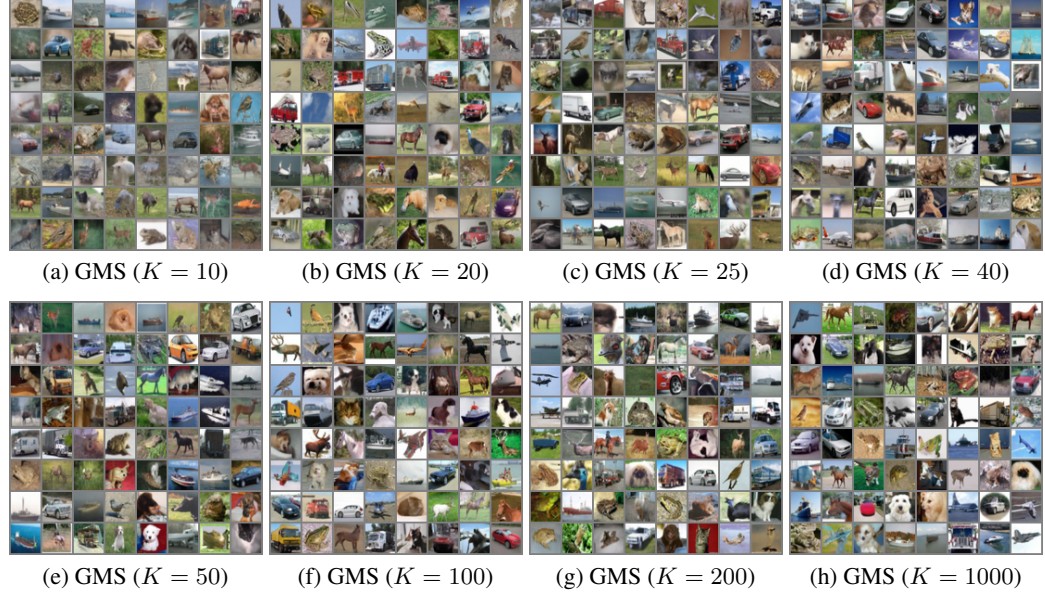

| (a) GMS ($K = 10$) | (b) GMS ($K = 20$) | (c) GMS ($K = 25$) | (d) GMS ($K = 40$) |

| (e) GMS ($K = 50$) | (f) GMS ($K = 100$) | (g) GMS ($K = 200$) | (h) GMS ($K = 1000$) |

Figure 10: Generated samples on CIFAR10 (LS)

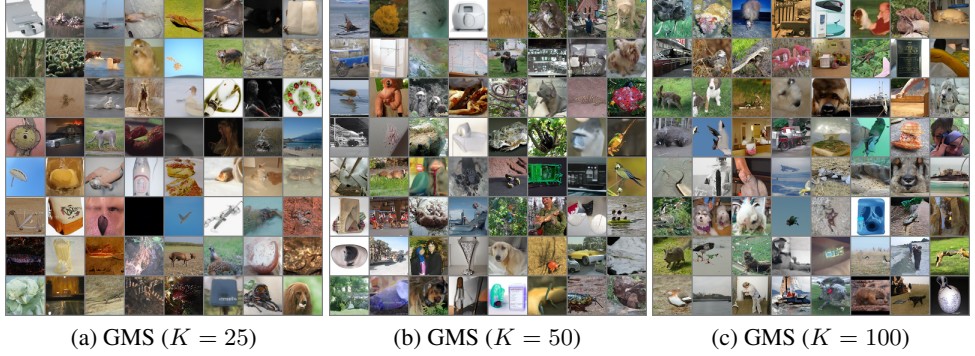

| (a) GMS ($K = 25$) | (b) GMS ($K = 50$) | (c) GMS ($K = 100$) |

Figure 11: Generated samples on ImageNet 64×64.

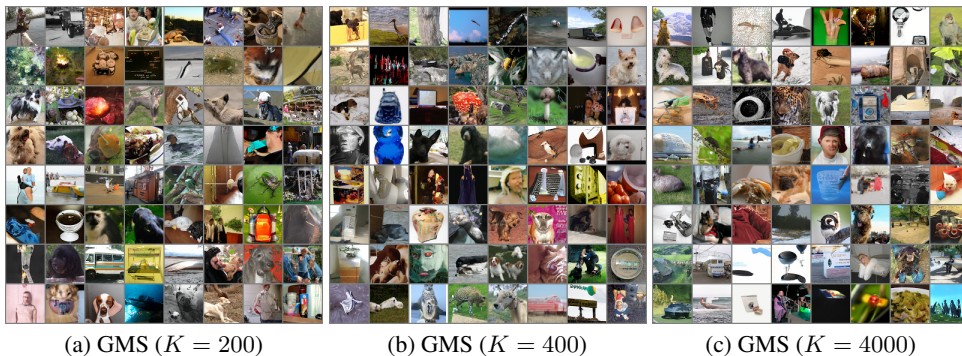

| (a) GMS ($K = 200$) | (b) GMS ($K = 400$) | (c) GMS ($K = 4000$) |

Figure 12: Generated samples on ImageNet 64×64.

