# OpenReview forum: "Gaussian Mixture Solvers for Diffusion Models"
_NeurIPS.cc/2023/Conference — NeurIPS 2023 poster_

### Official Review · Reviewer_iSEj · 2023-06-27

**Soundness:** 2 fair
**Presentation:** 1 poor
**Contribution:** 2 fair
**Rating:** 6
**Confidence:** 3

**Summary:**

The authors point out that $q(x_s|x_t)$ is not necessarily Gaussian when $t$ is significantly bigger than $s$, and propose to use a mixture of Gaussians in order to model better the reverse process, when the number of integration steps is not large. Such a selection guarantees that with an increasing number of steps, the mixture of Gaussians will subsequently adopt the form of the Gaussian. In order to estimate the parameters of the Gaussians constituting the mixture, the authors estimate the diagonals of higher order moments of the mixture through higher order denoising and integrate them in the generalized method of moments in order to estimate the desired parameters.

**Strengths:**

The paper is well motivated, as $q(x_s|x_t)$ is not necessarily Gaussian when $t$ is significantly bigger than $s$.

The experimental results show some improvements when the same number of steps are used.

**Weaknesses:**

**Main Weaknesses:**

1) The paper's presentation is not apt for publication.

The manuscript contains several sentences that are notably difficult to interpret. For instance, the authors' intent remains unclear in the sentence beginning with 'These modeling...' in line 158 and concluding on 160. This aspect only gains clarity in Equation 9, which is located in a different section.

A multitude of vital elements are left undefined. For instance, the parameters $\theta$ and $\theta^*$ are not properly concretized in the Gaussian Mixture context, and furthermore, the function $h$ in algorithm 2 lacks a definition in the main manuscript and Appendix B.

The paper's structure is counter-intuitive. In Algorithm 2, the authors initially compute the moments of the Gaussian mixture (a), and subsequently utilize them to calculate the mixture's defining parameters (b). Contrarily, in Section 3, the authors commence with (b) and proceed to (a). Since this order is not natural it makes the paper difficult to read, and the situation is exacerbated since at this point in a) all the definitions of parameters and moments are general and do not relate to the problem at hand. The connection between these general definitions and the specific problem tackled in the paper is never explicitly stated. It is important to show the connection between the parameters $\theta$, moments $M_n^{(GM)}(\theta)$, $M_n(x_i)$ on one hand, and $\pi_k, \mu_k, \sigma_k, \hat{M}_n$ on the other. This would enable writing the loss in Equation 8 explicitly, which is not done.

2) Sampling times (in seconds) are not provided.

 Considering that the premise of paper is enabling generation in fewer steps (that is, speeding up the sampling process), the sampling time curves of the sampling process for different number of steps should be provided in the main paper (Close to Table 1), for both the proposed method as well as the benchmarks. That is, a figure such as Figure 5 in Appendix E6 should be extended for numbers of steps above 50, and placed in the main paper. The values of the curves presented should be in seconds, instead of presenting the ratio between methods. Furthermore, it would be interesting to compare against the stochastic sampler in [1].

**Minor Weaknesses:**

1) Equation 9 is not derived in Appendix B.

2) Using a mixture of only two Gaussians is a bit underwhelming considering all the material.

3) The authors state that the "higher-order correlation between two pixels is assumed to be zero". While it is true that this premise aligns with the work of Bao et al., the authors should justify this in the paper.

4) The Figure in the introduction is misplaced. It does not elucidate the method. It presents results of the performance of the method, hence it should either go to the Experimental section or to the Appendix.

I am willing to increase my score if the authors extend Figure 5 (e.g. 100 200 1000 as in Table 1), and are willing to follow the presentation advices provided in Main Concern 1. Also there are grammar/spelling mistakes in the paper and the authors should ensure they are corrected.

[1] Karraset al, Elucidating the Design Space of Diffusion-Based Generative Models, NeurIPS 2022.

**Questions:**

What are the advantages of GMS in the light of recent methods such as consistency models [2]?

Have the authors performed experiments to measure how the sampling time per step increases when the number of Gaussians in the mixture (number of moments estimated) increases?

[2] Song et al, Consistency Models, ICML 2023


**Limitations:**

While limitations are presented, the authors could elaborate more on this topic.

---

> ### Author Rebuttal · Authors · 2023-08-10
>
> Thank you for your supportive review and suggestions.
>
> ***Main Weakness 1: Presentation***
>
> We appreciate your careful reading of our paper and your helpful suggestions. We will address these issues in the final version, including but not limited to:
>
> 1. We will revise the sentence in line 158 into "With such a network, the moments under the $q(x\_s|x\_t)$ measure can be decomposed into moments under the $q(x\_0|x\_t)$ measure, so that sampling any $x\_t$ to $x\_s$ requires only a network whose inputs are $x\_t$ and $t$." to make it clearer.
> 2. The variable $\\theta\^*$ means the potential optimal parameters for Gaussian mixture models given the first-to-third order of moments and the $\theta$ represents the variables required to optimize, which are denoted as $\mu^{(1)}_t$, $\mu^{(2)}_t$ and $\sigma^2_t$ in backward transition kernel $p(x_s|x_t) = \frac{1}{3}\mathcal{N}({\mu^{(1)}_t},{\sigma^2_t})+ \frac{2}{3}\mathcal{N}({\mu^{(2)}_t},{\sigma^2_t})$.
> 3. The function $h$ is defined as $h(f\^3\_{[1]}(x\_t,t),f\^3\_{[2]}(x\_t,t),f\^3\_{[3]}(x\_t,t))=M\_1(f\^3\_{[1]}(x\_t,t)),M\_2(f\^3\_{[2]}(x\_t,t)),M\_3(f\^3\_{[3]}(x\_t,t))$, where $M\_1(f)=E[x]$ in Eq.(22), $M\_2(f)=Cov(x)$ in Eq.(23) and $M\_3(f)=Ske(x)$ in Eq.(24).
> 4. We will re-organize the paper, explicitly state the connection between these general definitions and the specific problem tackled and the loss in Equation 8.
>
> We will also thoroughly correct the grammar/spelling mistakes in our paper.
>
> ***Main Weakness 2a: Sampling times (in seconds) are not provided***
>
> We have included the sampling time curves of the sampling process for different numbers of steps in Fig. C in the rebuttal PDF. Under the same sampling steps, our GMS incurs approximately 10% additional time compared to SN-DDPM due to extra optimization for Gaussian mixture solving. This optimization, however, contributes to improved quality, as shown in Table 1. Besides, we show the optimizing time curve which is required in GMS by changing the batch size and optimizing steps in Fig. C.
>
> For a more intuitive comparison, we also plot the FID curve w.r.t. the sampling time (in seconds) in Fig. A of the rebuttal PDF. Across the same computational budget (sampling time in seconds), we consistently observe improved performance over SN-DDPM.
>
> We will include these figures and results in the revision.
>
>
> ***Main Weakness 2b: Comparison with more baselines***
>
> Following your suggestions, we compare GMS with two additional SDE-based solvers: EDM [a] and SEED [b], with the latter being a higher-order SDE solver. We note that EDM in [a] use $x\_0$ prediction networks, which differ from ours and are thus not directly comparable. The symbol $\^*$ indicates network distinctions from the noise network in discrete time.
>
> | Method/NFE | 10 | 20 | 40 | 50 | 100 |
> | :---------------: | :---: |  :----:  | :----: |  :----: |  :----: |
> | GMS | 17.43 | 7.18 | 4.52 | 4.16  |  3.26 |  3.01 | 2.76 |
> | EDM[a]$\^*$ | 35.07 | 14.04 | 9.56 | 5.12 | 2.99  |   |
> | SEED-2[b] | 481.09 | 305.88 | 51.41 | 11.10 |  3.19 |
> | SEED-3[b] | 483.04 | 462.61 | 247.44| 62.62 | 3.53 | 3.08 |
>
> We observe that our GMS consistently outperforms within 50 NFEs. We will add these result in the revision.
>
> ***Minor Weakness 1: Derivation about Eq. (9)***
>
> Thanks for the comments. We will derive Eq. (9) step by step in Appendix B in the final version.
>
>
> ***Minor Weakness 2: Reasons for using a mixture of only two Gaussians***
>
> It's worth noting that in GMS, we use a mixture of two Gaussians (i.e., a two-mode distribution) to effectively capture the reverse transition kernels per sampling step. Intuitively, this choice can potentially encompass exponential modes across the entire trajectory. Empirically, we consistently observe that employing a mixture of two Gaussians yields favorable results across all settings in our experiments.
>
>
> ***Minor Weakness 3: Assumption of zero higher-order correlation***
>
> Estimating full higher-order moments results escalated output dimensions (e.g., quadratic growth for covariance and cubic for the third-order moments) and thus requires substantial computational demands. We therefore consider the diagonal higher-order moments in our method for computational efficiency, similar to Bao et al. [c]. We will clarify this in the revision.
>
>
>
> ***Minor Weakness 4: The placement and the function of Figure 1***
>
> The primary purpose of including Fig. 1 in the introduction was to provide readers with a visual insight into the motivation behind our method. The figure showcases the diminishing effectiveness of Gaussian reverse transition kernels (SN-DDPM) when using fewer discretization steps, which motivates the choice of non-Gaussian transition kernel in this paper. We are also open to further discussion on this issue and willing to re-organize it for better presentation.
>
>
> ***Question 1: Consistency models***
>
> Consistency models [d] focus on distilled ODE-based solvers, while our work is an SDE-based solver. Notably, SDE-based solvers offer distinct advantages in various downstream tasks, including stroke-based synthesis, image translation, and image manipulation, as outlined in Sec. 1. We will discuss the relation with [d] in the final version.
>
>
> ***Question 2: How does the sampling time per step increase when the number of Gaussians in the mixture increases?***
>
> In our experiments, we did not try increasing the number of Gaussians in the mixture (as well as the number of moments estimated) due to the reason mentioned in our response to ***Minor Weakness 2***.
>
> ***References***
>
> [a] Karras et al. Elucidating the Design Space of Diffusion-based Generative Models. 2022\
> [b] Gonzalez et al. SEEDS: Exponential SDE Solvers for Fast High-Quality Sampling from Diffusion Models. 2023\
> [c] Bao et al. Estimating the Optimal Covariance with Imperfect Mean in Diffusion Probabilistic Models. 2022\
> [d] Song et al. Consistency Models. 2023

---

> > ### Comment · Reviewer_iSEj · 2023-08-11
> >
> > Thank you for your response. My main concerns have been addressed and hence I adjusted the score accordingly.
> > As you intend to discuss the relation with [d] in your final version, it is important to note that consistency models trained in isolation are not focused on distilled ODE-based solvers (Theorem 2 and Algorithm 3).

---

> > > ### Author Response · Authors · 2023-08-11
> > > **Thank you for your feedback**
> > >
> > > Thank you for your detailed comments. We will discuss the relation between our work and consistency models [d] trained both via distillation (i.e., CD) and in isolation (i.e., CT) in the final version. We appreciate your time and effort in reviewing our paper. Thank you!

---

### Official Review · Reviewer_iDhp · 2023-06-27

**Soundness:** 2 fair
**Presentation:** 3 good
**Contribution:** 3 good
**Rating:** 6
**Confidence:** 3

**Summary:**

The authors address the efficiency-effectiveness dilemma faced by existing SDE-based solvers in diffusion models during inference. They observe that the Gaussian assumption in the reverse transition kernel is frequently violated, even with a limited number of discretization steps. To overcome this limitation, the authors propose a new class of SDE-based solvers called Gaussian Mixture Solvers (GMS). In this approach, they estimate the first three-order moments and optimize the parameters of a Gaussian mixture transition kernel using generalized methods of moments. They present empirical results that demonstrate that GMS outperforms other SDE-based solvers in terms of sample quality for image generation and stroke-based synthesis in various diffusion models.

**Strengths:**

The authors provide a solid theoretical foundation for their approach, highlighting the discrepancy between the empirical data and the assumptions made for the Gaussian transition kernel commonly used in models like SN-DDPM. They tackle this challenge by proposing a novel and effective solution tailored to address this specific issue. Their innovative approach not only takes into account theoretical benefits but also considers computational constraints.

**Weaknesses:**

Despite introducing GMS as a potential resolution to the efficiency-effectiveness dilemma, the authors fail to provide compelling evidence in their presentation to substantiate their assertion.

**Questions:**

1. Can you provide further clarification regarding the claim made in the abstract and various sections of the text that a limited number of discretization steps amplifies the violation of assumptions for the Gaussian transition kernel used in SN-DDPM? How does this relate to the performance of GMS compared to SN-DDPM when considering a small number (e.g., 100) versus a larger number (e.g., 1000) of discretization steps?

2. In Table 1 and Figure 3, it is apparent that the relative (or absolute) improvement of GMS over SN-DDPM is not significantly different when comparing a small number of discretization steps to a larger number. Can you address this discrepancy and explain why the expected noticeable improvement in GMS performance for a smaller number of discretization steps is not observed?

3. The authors mention that GMS is superior to SN-DDPM when considering computational cost (line 273). Could you provide more clarity on this claim and its implications? If GMS does indeed offer computational advantages, please elaborate on this aspect in the text and present corresponding results. Conversely, if this claim is not supported by evidence, please clearly address this discrepancy.

4. Considering the theoretical value of this work, a crucial question arises: Why would one choose to utilize GMS if comparable results can be obtained within the same computational budget using a simpler method like SN-DDPM? Please provide a compelling rationale for using GMS, taking into account its potential advantages and drawbacks compared to alternative approaches.

**Limitations:**

The authors emphasize that a limited number of discretization steps exacerbates the violation of assumptions for the Gaussian transition kernel in SN-DDPM, as stated in the abstract and various sections of the text (e.g., line 39, 50, 117, and 290). However, it is surprising to note that the expected significant improvement of GMS over SN-DDPM for a smaller number (e.g., 100) versus a larger number (e.g., 1000) of discretization steps is not observed, as evident from table 1 and figure 3. This discrepancy requires clarification.

Additionally, the authors vaguely assert that GMS outperforms SN-DDPM when considering computational cost (line 273). If this claim holds true, it is crucial to provide clear explanations and present corresponding evidence in the text and results. Conversely, if this claim is not substantiated, it should be explicitly addressed. Although this work holds theoretical value, the fundamental question remains: What is the justification for using GMS if comparable results can be achieved within the same computational budget using a simpler method like SN-DDPM?

---

> ### Author Rebuttal · Authors · 2023-08-10
>
> Thank you for your valuable comments and questions.
>
>
> ***Question 1: Clarification regarding the claim that a limited number of discretization steps amplifies the violation of assumptions for the Gaussian transition kernel used in SN-DDPM***
>
> Thanks for the insightful comment. We clarify this issue and confirm our claim from both theoretical and empirical standpoints.
>
> Theoretically, we apply Bayes' rule to the posterior distribution $q(x_t|x_{t\~+\~\Delta~ t})$ as follows:
> $q(x_t|x_{t\~+\~\Delta~ t}) = \dfrac{q(x_{t\~+\~\Delta~ t}|x_{t})q(x_t)}{q(x_{t\~+\~\Delta~ t})} =q(x_{t\~+\~\Delta~ t}|x_{t})\exp(\log (q(x_t))-\log(q(x_{t\~+\~\Delta~ t})))$
> $\propto \exp(-\dfrac{\left \\| x_{t\~+\~\Delta~ t}-x_t-f_t(x_t)\Delta~ t \right \\|^2 }{2g_t^2\Delta~ t} +\log p(x_t)-\log(x_{t\~+\~\Delta~ t})),$
> where $\Delta~ t$ is the step size, $q(x_t)$ is the marginal distribution of $x_t$. When $x_{t\~+\~\Delta~ t}$ and $x_{t}$ are close enough, using Taylor expansion for $\log p(x_{t\~+\~\Delta~ t})$, we could obtain:
> $\log p(x_{t\~+\~\Delta~ t}) \approx \log p(x_t)+(x_{t\~+\~\Delta~ t}-x_t)\nabla_{x_t} \log p(x_t)+\Delta~ t \dfrac{\partial}{\partial t} \log p(x_t)$,
>
> $q(x_t|x_{t\~+\~\Delta~ t})\propto \exp(-\dfrac{\left \\| x_{t\~+\~\Delta~ t}-x_t-[f_t(x_t)-g_t^2\nabla_{x_t} \log p(x_t)]\Delta~ t \right \\|^2 }{2g_t^2\Delta~ t} +O(\Delta~ t))$.
> By ignoring the higher order terms, the reverse transition kernel will be Gaussian distribution. However, as $\Delta~ t$ increases, the higher-order terms in the Taylor expansion cannot be disregarded, which causes the reverse transition kernel to deviate from a Gaussian distribution. We will include these clarification in the revision.
>
> Empirically, from Fig. 2 in our paper, we observe that as the number of sampling steps decreases, the backward transition kernel increasingly deviates from a Gaussian distribution. For more details, please refer to Appendix D.2 in our paper.
>
> Please see our response to ***Question 2*** for analysis of the emprical performance in terms of FID.
>
>
> ***Question 2: About the improvement in a limited number of steps***
>
> As shown in Tab. 1 of the submission (we also plot a curve in Fig. B of the rebuttal PDF for a clearer illustration), GMS achieves an FID improvement of >4.0 with a small number of steps (e.g. 10) and <0.5 with a large number of steps (e.g. 200).
>
> However, we respectively clarify that due to the nonlinear nature of the FID metric, the relative/absolute FID improvements are not directly comparable across different number of steps. We will make it clearer in the final version.
>
>
> ***Question 3: About the claims that GMS is superior to SN-DDPM when considering computational cost***
>
> The intended statement here is: "(In the SDEdit task,) GMS exhibits superior performance compared to SN-DDPM **under the same** computational cost." We indeed observed a performance-time curve similar to that of the unconditional generation experiments (Fig. A in the rebuttal PDF), providing evidence for our intended claim.
>
> Thank you for pointing this out, and we will correct the original inaccurate claim.
>
>
> ***Question 4: Why choosing GMS if comparable results can be obtained within the same computational budget using a simpler method like SN-DDPM***
>
> We would like to clarify that GMS can achieve superior performance under the same computational budget. To provide an intuitive comparison, we present the CIFAR-10 FID curve against the computation time in Fig. A of the rebuttal PDF, where GMS consisently improves SN-DDPM when varying the computational budgets.
>
> Besides, we provide more results compared to other advanced SDE-based sovlers including EDM [b] and SEED [c], GMS still outperforms within limited number of discretization steps. See details in response to ***Main Weakness 2b*** of Reviewer iSEj.
>
>
> ***References***
>
> [a] Song et al. Score-based generative modeling through stochastic differential equations. 2020\
> [b] Karras et al. Elucidating the Design Space of Diffusion-based Generative Models. 2022\
> [c] Gonzalez et al. SEEDS: Exponential SDE Solvers for Fast High-Quality Sampling from Diffusion Models. 2023

---

> > ### Comment · Reviewer_iDhp · 2023-08-18
> >
> > Thanks for the rebuttal.
> >
> > The authors have address most of my concerns and hence I have adjusted the score.
> >
> > A notable lingering weakness pertains to whether the relatively modest enhancements can be justified by the heightened intricacy of the approach. This, indeed, warrants additional exploration.

---

> > > ### Author Response · Authors · 2023-08-18
> > > **Thank you for your feedback**
> > >
> > > Thank you for your feedback and for raising the score. We will discuss the limitation related to the added intricacy in the final version. We appreciate your time and effort in reviewing our paper. Thank you again!

---

### Official Review · Reviewer_iBkM · 2023-07-06

**Soundness:** 3 good
**Presentation:** 3 good
**Contribution:** 3 good
**Rating:** 5
**Confidence:** 4

**Summary:**

Sampling from diffusion models is equivalent to solving the reverse diffusion SDEs or the corresponding probability flow ODEs. In comparison, SDE-based solvers can generate samples of higher quality and are suited for image translation tasks. However, during inference, existing SDE-based solvers are severely constrained by the efficiency-effectiveness dilemma. To overcome this limitation, this paper introduces a novel class of SDE-based solvers called Gaussian Mixture Solvers (GMS) for diffusion models. Experimental results validate the motivation and effectiveness of GMS solvers.


---Post-rebuttal:
The authors address some of my concerns, However, I strongly enough the authors to conduct more experiments. I would like to keep my score.

**Strengths:**

a.	This paper systematically examines the assumption of the Gaussian transition kernel and reveal that it can be easily violated under a limited number of discretization steps even in the case of simple mixture data. To this end, the authors propose a new type of SDE-based solver called Gaussian Mixture Solvers.
b.	This paper presents an approach for estimating the high-order moments utilizing noise networks.


**Weaknesses:**

a. In Sec. 3.2, these authors claim that a designed Gaussian mixture model can degenerate to a Gaussian, however, it is not clear how to design such a Gaussian mixture model.
b. As the weight in Eq. 7 is manually setting, how to select an optimal set of weights for a specified testing benchmark?
c. In Table 1, except for the FID score, extra computational cost and running time should also be compared to verify the effectiveness of the proposed GMS solvers.
d. From Table 1, we can observe that the improvement of GMS is not so obvious when compared with SN-DDPM. Especially from Table 9 in the supplemental material, we see that with the same computation cost, the gap between GMS and SN-DDPM is small.
e. In the experiments, the resolution of the dataset is small, how about the improvement of GMS solvers on data with larger scale?
f. The experimental section is not enough.


**Questions:**

Please refer to the weakness

**Limitations:**

Yes

---

> ### Author Rebuttal · Authors · 2023-08-10
>
> Thank you for your supportive review and valuable comments.
>
>
> ***Weakness (a): Design of the Gaussian mixture model & How can it degrade to a Gaussian***
>
> Our design choice of the Gaussian mixture model for the reverse transition kernel is $p(x\_s|x\_t)=\\frac{1}{3} \\mathcal{N}(\\mu\^{(1)}\_t(x\_t),\\sigma\^2\_t(x\_t))+\\frac{2}{3} \\mathcal{N}(\\mu^{(2)}\_t(x\_t),\\sigma\^2\_t(x\_t))$ (as described in L182 of Sec. 3.2), where $\\mu\^{(1)}\_t(x\_t)$, $\\mu\^{(2)}\_t(x\_t)$ and $\\sigma\^2\_t(x\_t)$ are the three parameters to be optimized given the first three-order moments. It can degenerate to a Gaussian when $\\mu\^{(1)}\_t(x\_t)=\\mu\^{(2)}\_t(x\_t)$, in which case we have $p(x\_s|x\_t)=\\mathcal{N}(\\mu\^{(1)}\_t(x\_t),\\sigma\^2\_t(x\_t))$. We will clarify in the revision.
>
>
> ***Weakness (b): How to select an optimal set of weights?***
>
> For a Gaussian mixture with two components (i.e., our design choice), determining optimal component weights and parameters from only the first three-order moments is underdetermined. To tackle this, we pragmatically set the weights to ($\\frac{1}{3}$, $\\frac{2}{3}$) and observe that such choice consistently yielded superior performance among different datasets. Identifying an optimal weight value remains a promising direction for further enhancing the GMS. We will clarify in the revision.
>
>
> ***Weakness (c & d): Computational cost and running time should also be compared & The improvement when considering the extra computation time***
>
> Thank you for your suggestions. As our GMS enhances sample quality at the expense of additional computation, we provide further insights by presenting the CIFAR10 FID curve plotted against sampling time (in seconds) in Fig. A of the rebuttal PDF, which should be more intuitive than Table 9. Notably, a consistent and demonstrable improvement over SN-DDPM emerges within the same computational budget (sampling time in seconds), showing the superiority of our method. Specifically, we observe an FID improvement of \~0.9 (\~0.6, \~0.3, resp.) at sampling time of 40s (60s, 200s, resp.). We will incorporate this figure adjacent to Table 1 in the revision.
>
> Other analysis and results pertaining to computational cost and running time can be found in the Appendix. E.6. Specifically, Fig. 5 shows that GMS incurs approximately 10% higher computational time per step compared to SN-DDPM, while Fig. 6 offers a breakdown of time allocation across various components.
>
>
> ***Weakness (e): Experiments on large-scale data***
>
> Given the limited time available for rebuttal, we were unable to finish experiments on larger images. Nonetheless, we are actively engaged in experiments on ImageNet 256*256, and we are committed to delivering our best efforts in this regard.
>
>
> ***Weakness (f): The experimental section is not enough.***
>
> We have supplemented the following experiments and results to better demonstrate the effectiveness of GMS:
> - we present the CIFAR10 FID curve plotted against sampling time (in seconds) for SN-DDPM and GMS in Fig. A of the rebuttal PDF.
> - we present the sampling time (in seconds) curve plotted against the NFE (Number of Function Evaluation) for SN-DDPM and GMS in Fig. C of the rebuttal PDF.
> - we compare GMS with more SDE-based solvers such as EDM [a] and SEED [b]. Results show that GMS consistently outperforms within 50 NFEs. Please see details in response to ***Main Weakness 2b*** of ***Reviewer iSEj***.
>
>
> We are open to conducting additional experiments if they would contribute to a more comprehensive evaluation. Thank you for your feedback.
>
>
> ***References***
>
> [a] Karras et al. Elucidating the Design Space of Diffusion-based Generative Models. 2022\
> [b] Gonzalez et al. SEEDS: Exponential SDE Solvers for Fast High-Quality Sampling from Diffusion Models. 2023

---

> ### Author Response · Authors · 2023-08-21
> **Providing additional results for Reviewer iBkM**
>
> Thanks for your feedback. In this reply, we added the experiments of the large resolution (Imagenet 256$\\times$256). Combined with the content of the first rebuttal, we supplemented all the experiments according to your comments. In particular, we chose one of the SOTA latent diffusion models in Imagenet 256$\\times$256, called U-ViT; we use the U-ViT-Huge from [c] as our backbone network and train extra the second-order and the third-order noise prediction heads with two transformer blocks with the frozen backbone (details in line 226). With the same sampling parameters such as the total denoising time steps and values of cfg, the conclusions of the experiments remain unchanged, GMS outperforms SN-DDPM within the same number of steps.
>
> FID results of class-conditional image generation on ImageNet 256$\\times$256:
> | Method/# Steps | 15 | 20 | 25 | 40 | 50 | 100 | 200 |
> | :-: | :-: |:-: |  :-:  | :-: |  :-: |  :-: | :-: |
> | DDPM | 6.48 | 5.30 | 4.86 | 4.42 | 4.27 | 3.93 | 3.32 |
> | SNDDPM | 4.40 | 3.36 | 3.10 | 2.99 | 2.97 | 2.93 | 2.82 |
> | GMS |4.01 | 3.07 | 2.89 | 2.88 | 2.85 | 2.81 | 2.74 |
>
> FID results of unconditional image generation on ImageNet 256$\\times$256:
> | Method/# Steps | 25 | 40 | 50 | 100 |
> | :-: | :-:  | :-: |  :-: |  :-: |
> | DDPM |  8.62 | 6.47 | 5.97 | 5.04 |
> | SNDDPM |  8.19 | 5.73 | 5.32| 4.60 |
> | GMS | 7.78 | 5.42 | 5.03| 4.45 |
>
> ***References***
>
> [c] Bao et al. "All are worth words: A vit backbone for diffusion models." Proceedings of the IEEE/CVF Conference on Computer Vision and Pattern Recognition. 2023.

---

### Official Review · Reviewer_naDW · 2023-07-06

**Soundness:** 3 good
**Presentation:** 2 fair
**Contribution:** 2 fair
**Rating:** 7
**Confidence:** 3

**Summary:**

The paper proposes to weaken the Gaussian assumption of the transition probability in the reverse SDE used in deep diffusion models. They first illustrates how and when the Gaussian assumption is wrong. Then they suggest to approximate the non-Gaussian transition probability by a Gaussian Mixture which is adjusted with the method of moments. Finally, they illustrates the superiority of their methods on different standard datasets (CIFAR, ImageNet64) using standard metric (FID).

**Strengths:**

- The manuscript is well-written and the problem is clearly stated and illustrated.
- The proposed method allows to reduce the number of time step
- The results are superior in terms of sample quality (measured by FID).

Specifically appreciated :
- The authors account for the computational cost of their method in the supp material.
- The authors are transparent about real-time applicability of their methods.

**Weaknesses:**

- It's unclear what is the interest of the proposed method beyond mathematical and empirical curiosity.
- The proposed method weakens an assumption at an extra-computational cost. When accounting for this extra-cost improvements are lowered.


**Questions:**

I did not fully understand why the method of moments is preferred. To me, the issue of estimating a mixture model that depends on both s and t also apply to the estimate of moment. What is the advantage of using the method of moments ?
EM algorithm is easy to implement and I feel like you might spare computational resources compared to the calculation of higher-order moment. Have you try EM ?

Fig 1 : You should plot  the log density to make more visible the differences between the two curves.

**Limitations:**

ok

---

> ### Author Rebuttal · Authors · 2023-08-10
>
> Thank you for your supportive review and valuable suggestions.
>
>
> ***Weakness 1: The interest of the proposed method***
>
> We understand that your concern may relate to the theoretical contributions and practical benefits of our method. (We acknowledge the possibility of potential misunderstanding and welcome any further questions.)
>
> We find that the Gaussian assumption in the reverse transition kernel can be violated, as shown by our theory and experiments (Fig. 1). Building on this, we introduce the GMS. This new approach improves the efficiency of SDE-based solvers.
>
>
> SDE-based solvers have notable advantages over ODE-based solvers in various downstream applications. Noteworthy examples include stroke-based synthesis, expounded upon in Section 4.2, and image translation, as detailed in [a]. These instances underscore the superior efficacy of SDE-based solvers. Concurrently, when an ample quantity of sampling steps is employed, the performance of SDE-based solvers exhibits superior outcomes in both unconditional and conditional sampling scenarios (as demonstrated in [b]).
>
> Our experiments in Fig. A of the rebuttal PDF show that **GMS achieves superior performacne than competitive SDE-solvers under the same computational budget**. Therefore, we believe that GMS is promising in applications investigated in [a,b].
>
>
> ***Weakness 2: Extra-cost during inference***
>
> Our GMS achieves superior performance under the same computational budget. To provide an intuitive comparison, we present the CIFAR10 FID curve plotted against sampling time (in seconds) in Fig. A of the rebuttal PDF. Notably, a consistent improvement margin over SN-DDPM emerges within the same computational budget (sampling time in seconds). Specifically, we observe an FID improvement of \~0.9 (\~0.6, \~0.3, resp.) at sampling time of 40s (60s, 200s, resp.). This observation corroborates our claim that the Gaussian reverse transition kernel deviates when employing fewer discretization steps. We will clarify this in the revision.
>
>
> ***Question 1: Advantages of the method of moments compared to the EM algorithm***
>
> The EM algorithm is not well suited for our method due to the following reasons:
> 1. Nontrivial training and loss modifications: Learning the reverse transition kernel $p(x\_s|x\_t)=\\sum\_{i=1}\^{M}w\_i \\mathcal{N}(x\_s|{\\mu\_i}(x\_t),{\\Sigma\_i}(x\_t))$ ($\\sum\_iw\_i=1$) with EM algorithm requires alternatively performing expectation and maximization steps, where the latter are not differentiable. Moreover, it requires sampling time step pairs $(s, t)$ and many-to-one pairs of $x\_s$ and $x\_t$, incurring computational costs. In contrast, our approach seamlessly expands with the noise prediction loss within the diffusion framework and maintains (training) efficiency.
> 2. Substantial model architecture changes: EM necessitates nontrivial architectural changes for handling double time inputs $(s, t)$ in modeling $p(x\_s|x\_t)$. This poses higher model capacity demands and complex design challenges.
>
> We will include these in the revision.
>
>
> ***Question 2: Re-plot Figure 1***
>
> Thank you for your valuable suggestion. We will update it in the final revision.
>
>
> ***References***
>
> [a] Zhao et al. Egsde: Unpaired image-to-image translation via energy-guided stochastic differential equations. 2022\
> [b] Karras et al. Elucidating the Design Space of Diffusion-based Generative Models. 2022

---

> > ### Comment · Reviewer_naDW · 2023-08-15
> > **post-rebuttal response**
> >
> > Thanks for the feedback.
> >
> > Why do you say that the M-step is not differentiable in Gaussian Mixture Models ? I am not sure of that because the M-step in GMM is in closed form so it feels like it could be differentiable, no ?

---

> > > ### Author Response · Authors · 2023-08-16
> > > **Thank you for your feedback**
> > >
> > > Thank you for your feedback. We appreciate the opportunity to provide more detailed clarification on the challenge of utilizing the EM algorithm to learn the reverse transition kernel.
> > >
> > > Let $p(z\_s=i|x\_s,x\_t)$ denote the posterior probability that the point $x\_s$ belongs to mixture component $i$, where $i\\in\\{1,\\cdots,M\\}$. In general, parameterizing the mixture distributions $p(x\_s|z\_s=i,x\_t), i=1,\\cdots,M$ with neural networks will lead to an intractable $p(z\_s=i|x\_s,x\_t)$ and challenges in the expectation steps. To tackle this, common approaches like Monte Carlo EM approximate the expectation steps through sampling from $p(z\_s|x\_s,x\_t)$, often involving non-differentiable processes like MCMC sampling. The non-differentiability of these samples can hinder a fully differentiable learning process, necessitating iterative updates.
> > >
> > > However, our specific case benefits from the Gaussian form of $p(x\_s|z\_s=i,x\_t)$, making both the expectation and maximization steps tractable and differentiable. We will properly address this in the revision. Nevertheless, the need for paired samples, required loss modifications (rather than the simple noise prediction), and potential nontrivial architectural adjustments remain key obstacles to the seamless integration of the EM algorithm into our method.

---

> > > > ### Comment · Reviewer_naDW · 2023-08-18
> > > >
> > > > Thanks for the answer. I am indeed not requiring you to implement the EM version for the current submission. Yet, It must be discussed and it would be great to add some more details in the appendix about the changes that would be required for EM to work.
> > > >
> > > > I have adjusted my rating.

---

> > > > > ### Author Response · Authors · 2023-08-18
> > > > > **Thank you for the additional comments**
> > > > >
> > > > > Thank you for your additional comments and for raising the score. We will incorporate all the clarifications and new results into the final version of the paper. Your time and effort in reviewing our work are greatly appreciated. Thank you again!

---

### Author Rebuttal · Authors · 2023-08-10

We thank all reviewers for their valuable and constructive feedback, and we have responded to each reviewer individually. We have also uploaded a rebuttal PDF that includes:
- **Fig. A**: The relation between the sample quality (in FID) and the sampling time (in seconds) of GMS and SN-DDPM on CIFAR10.
- **Fig. B**: The reduction in the FID on CIFAR10 for GMS compared to SN-DDPM when sampling with different number of steps.
- **Fig. C**: The relation between the sampling/optimizing times (in seconds) and the sampling/optimizing steps of GMS on CIFAR10.

In addition, we have extended our experimental comparisons to include more baselines, such as EDM from [a] and SEED from [b].

***References***

[a] Karras et al. Elucidating the Design Space of Diffusion-based Generative Models. 2022\
[b] Gonzalez et al. SEEDS: Exponential SDE Solvers for Fast High-Quality Sampling from Diffusion Models. 2023

---

### Decision · Program_Chairs · 2023-09-21

**Decision:**

Accept (poster)

**Comment:**

The reviewers appreciated the methodological contribution addressing the issue with Gaussian transition kernel in diffusion models, while emphasising the clarity of the presentation and the thorough discussion of computational contraints. Questions addressed several aspects of this work, among which the need for clarifications on methodological and computational aspects, while requiring additional experimental evidence.

The rebuttal and subsequent discussion were successful in addressing these questions and, as result, reviewers either confirmed or increased their score towards acceptance. Overall, the paper presents a relevant contribution to the conference.